# ROYAL SOCIETY
# OPEN SCIENCE

Science, society and policy

palaeontology/palaeontology/taxonomy and systematics

scientific colonialism, parachute science, research ethics, palaeontological heritage, illegal fossil trade, Latin America

**Authors for correspondence:**
Juan Carlos Cisneros
e-mail: juan.cisneros@ufpi.edu.br
Emma M. Dunne
e-mail: dunne.emma.m@gmail.com

# Digging deeper into colonial palaeontological practices in modern day Mexico and Brazil

Juan Carlos Cisneros[1], Nussaïbah B. Raja[2], Aline M. Ghilardi[3], Emma M. Dunne[4], Felipe L. Pinheiro[5], Omar Rafael Regalado Fernández[6], Marcos A. F. Sales[7], Rubén A. Rodríguez-de la Rosa[8], Adriana Y. Miranda-Martínez[9], Sergio González-Mora[10], Renan A. M. Bantim[11], Flaviana J. de Lima[12] and Jason D. Pardo[13]

[1]Museu de Arqueologia e Paleontologia, Universidade Federal do Piauí (UFPI), Teresina, PI 64049-550, Brazil
[2]GeoZentrum Nordbayern, Department of Geography and Geosciences, Friedrich-Alexander University Erlangen-Nürnberg, Loewenichstr. 28, 91054 Erlangen, Germany
[3]Departamento de Geologia, Universidade Federal do Rio Grande do Norte (UFRN), Natal, Brazil
[4]School of Geography, Earth and Environmental Sciences, University of Birmingham, Edgbaston, Birmingham, B15 2TT, UK
[5]Laboratório de Paleobiologia, Universidade Federal do Pampa, São Gabriel, Brazil
[6]Mathematisch-Naturwissenschaftliche Fakultät der Universität Tübingen Fachbereich Geowissenschaften, Tübingen, Germany
[7]Instituto Federal de Educação, Ciência e Tecnologia do Ceará (IFCE) – Campus Acopiara, Acopiara, Ceará, Brazil
[8]Unidad Académica de Ciencias Biológicas-Unidad Académica de Ciencias de la Tierra, Universidad Autónoma de Zacatecas, Calzada Solidaridad, S/N, Campus II, C.P. 98060, Zacatecas, Mexico
[9]Departamento de Biología Evolutiva, Facultad de Ciencias, Universidad Nacional Autónoma de México, Ciudad Universitaria, 04510 Ciudad de México, México
[10]Museo de Paleontología, Departamento de Biología Evolutiva, Facultad de Ciencias, Universidad Nacional Autónoma de México, Ciudad Universitaria, 04510 Ciudad de México, México
[11]Laboratório de Paleontologia, Departamento de Ciências Biológicas, Universidade Regional do Cariri, Rua Coronel Antônio Luís, 1161, Pimenta, Crato, Ceará, Brazil
[12]Laboratório de Paleobiologia e Microestruturas, Centro Acadêmico de Vitória – Universidade Federal de Pernambuco (CAV/UFPE), R. Alto do Reservatório – Alto José Leal, Vitória de Santo Antão, Pernambuco, Brazil
[13]University of Calgary, Calgary, Alberta, Canada T2N 4N1

 JCC, 0000-0001-6159-1981; NBR, 0000-0002-0000-3944;
AMG, 0000-0001-9136-0236; EMD, 0000-0002-4989-5904;
FLP, 0000-0003-3354-914X; ORRF, 0000-0002-6247-6181;
MAFS, 0000-0002-2292-578X; RAR-dlR, 0000-0002-7219-1550
SG-M, 0000-0001-9709-2033; RAMB, 0000-0003-4576-0989;
FJdL, 0000-0001-8602-6508; JDP, 0000-0002-2665-8893

Scientific practices stemming from colonialism, whereby middle- and low-income countries supply data for high-income countries and the contributions of local expertise are devalued, are still prevalent today in the field of palaeontology. In response to these unjust practices, countries such as Mexico and Brazil adopted protective laws and regulations during the twentieth century to preserve their palaeontological heritage. However, scientific colonialism is still reflected in many publications describing fossil specimens recovered from these countries. Here, we present examples of 'palaeontological colonialism' from publications on Jurassic–Cretaceous fossils from NE Mexico and NE Brazil spanning the last three decades. Common issues that we identified in these publications are the absence of both fieldwork and export permit declarations and the lack of local experts among authorships. In Mexico, access to many fossil specimens is restricted on account of these specimens being housed in private collections, whereas a high number of studies on Brazilian fossils are based on specimens illegally reposited in foreign collections, particularly in Germany and Japan. Finally, we outline and discuss the wider academic and social impacts of these research practices, and propose exhaustive recommendations to scientists, journals, museums, research institutions and government and funding agencies in order to overcome these practices.

## 1. Introduction

Scientific advances played an important role in supporting the agenda of European colonialism. Scientific curiosity is noted as one of the key motivations behind the expeditions that led to the colonization and annexation of regions in Asia, Africa and the Americas [1]. As a result, many 'exotic' specimens collected by naturalists or geologists from colonies were sent back to the respective colonial state, to adorn houses of high-ranked members of society or to be reposited in national scientific societies or institutions, with the purpose of scientific inquiry [1,2]. The latter led to the establishment of large museums to house the vast collections of curiosities brought back to Europe from overseas expeditions as well as imperial conquests within Europe. Although colonialism is frequently described in political, social and military contexts, it is also present in many scientific practices still in use today. The development of scientific disciplines, educational programmes and academic organizations were all products designed to benefit colonial advancement [3], e.g. advancements in geological tools allowed colonial powers to uncover and exploit several natural resources in colonies.

This structure of colonial science—derived from the practice of science in the colonies—has given rise to 'scientific colonialism' in the post-colonial world, some of whose extractive aspects are sometimes referred to as parachute science [4–7], helicopter research [8,9] or even parasitic science [7]. Within scientific colonialism, middle- and low-income countries are perceived as suppliers of data and specimens for the high-income ones, the contributions of local collaborators are devalued or omitted, and the legal frameworks in lower income countries are trivialized or even ignored [6,10–14]. In turn, colonialist nations owe their wealth to these extractive colonial practices that have existed for centuries, allowing them to accumulate knowledge, power and financial resources. These extractive practices persist in the field of palaeontology to this day [13,14].

In response to a long history of colonial science practices, many countries, most notably several Latin American countries, adopted protective laws and regulations in the twentieth century to preserve and protect their biological, archaeological and palaeontological heritage. Under these laws, fossils are considered to be property of the nation state, and their sale, purchase and permanent exportation is prohibited [15–17]. Countries like Brazil [18], Argentina [15], Colombia [19] and Chile [20] also make it compulsory for a foreign party to be associated with a local institution in order to conduct fieldwork in the respective country and collect fossil samples. Brazil and Mexico prohibit the commercial trade of their fossils, require permits for their temporary export and, in the case of Brazil, the permanent export of specimens used for describing new species (holotypes) is not allowed [21,22]. These two countries, despite the presence of laws and regulations, still fall victim to scientific colonialism, including the illicit trafficking of fossil specimens. In fact, the illegal trade of fossils in Brazil has been blamed on the presence of laws; Martin [23,24] states that the very fact that laws exist for the protection of these fossils could be the reason that officials can be bribed and these fossils can be sold for a considerable price on the illicit market.

Both Mexico and Brazil are former European colonies with vast territories, large sedimentary basins and a huge palaeontological potential that remains relatively unexplored. These characteristics, together with the predominance of an overall low-income population and weak local currencies, make them

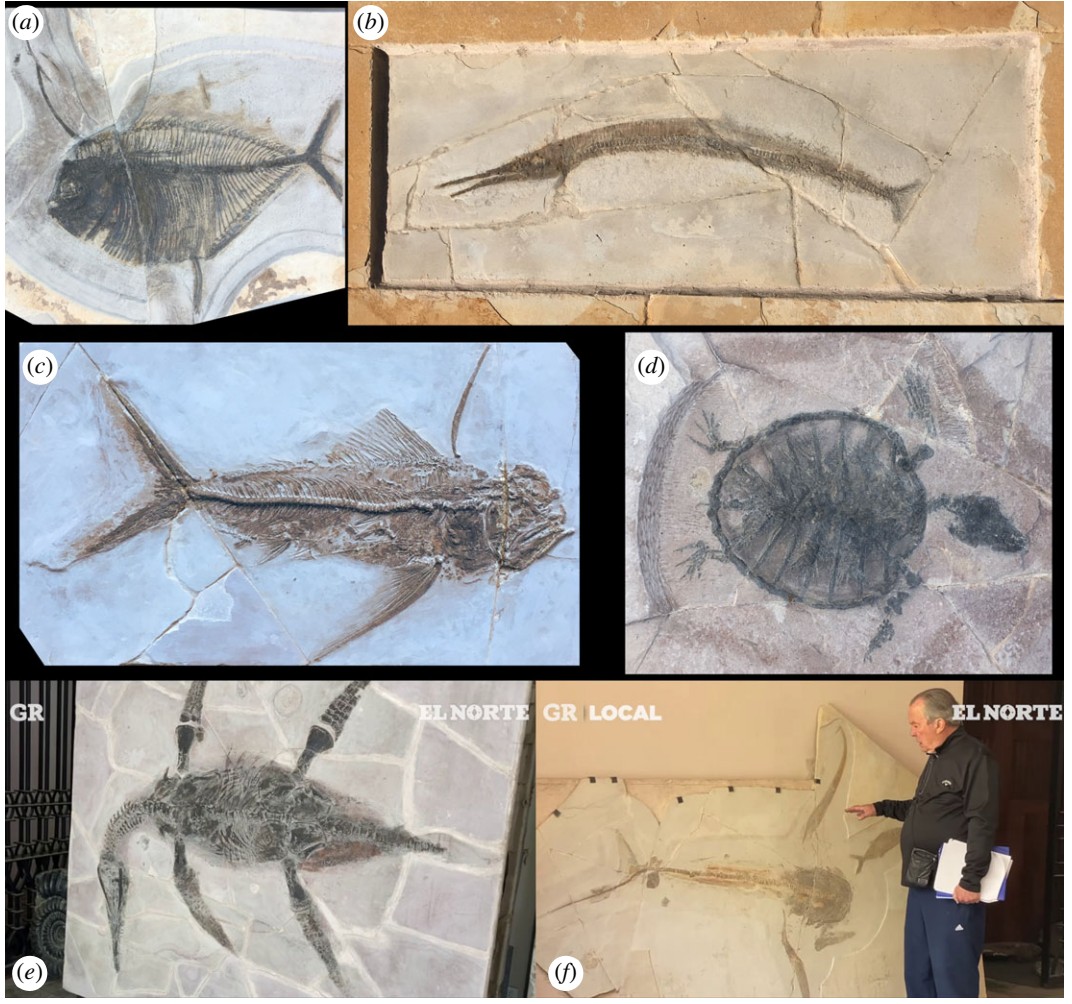

**Figure 1.** Fossils from the Sabinas Basin, Mexico, stored in a private collection. (*a*) Cf. *Tselfatia formosa*, approximately 750 mm body length. (*b*) Cf. *Belenostomus longirostris*, approximately 700 mm body length. (*c*) Pachyrhizodontid fish approximately 750 mm body length. (*d*) Chelonia cf. *Terlinguachelys* sp., approximately 300 mm body length. (*e*) Holotype of plesiosaur *Mauriciosaurus fernandezi* [25]. (*f*) Holotype of *Aquilolamna milarcae* [32]. All fossils are deposited in the collection registered by INAH as REG2544PF, which is housed by Mauricio Fernández (seen in the photograph) in Monterrey, Nuevo León, Mexico. (*e,f*) Image captures from video by Grupo Reforma Youtube Channel [31].

attractive targets for palaeontological colonialism. In the last decades, the Crato and Romualdo formations of the Araripe Basin in NE Brazil and the Sabinas, Parras and La Popa basins (Mexican Gulf) in NE Mexico have produced an unprecedented wealth of Jurassic to Cretaceous (200–66 million years ago) fossils. These extremely rich deposits of well-preserved fossils, known as *Lagerstätten*, have enriched our view of evolution, revealing a plethora of new vertebrates (figures 1 and 2), invertebrates, plants and fungi [34–37]. These exposures yield tantalizing examples of fossil preservation, including several soft-tissue instances [25,38–41]. Most of the published research output on fossils from these regions, however, has been led by foreign palaeontologists with the limited involvement of local researchers. Many of these studies are based on fossils that have been unethically and/or irregularly acquired and/or exported [42]. Several published fossils lack contextual geographical and geological information, while many important specimens are in private or foreign collections, where they can be difficult to access. Recent publications describing new fossil species such as the plesiosaur *Mauriciosaurus fernandezi* [25] (figure 1*e*) and the shark *Aquilolamna milarcae* [32] (figure 1*f*) from the Agua Nueva Formation (Sabinas Basin), Mexico as well as the snake-like reptile *Tetrapodophis amplectus* [27] (figure 2*f*) and the dinosaur '*Ubirajara jubatus*' [26] (figure 2*a*) both from the Crato Formation (Araripe Basin) Brazil, have raised a number of questions involving ethics, legal issues and scientific reproducibility. In this study, we present and discuss the academic and social impact of the research published during 1990–2021 that represents examples of scientific colonialism

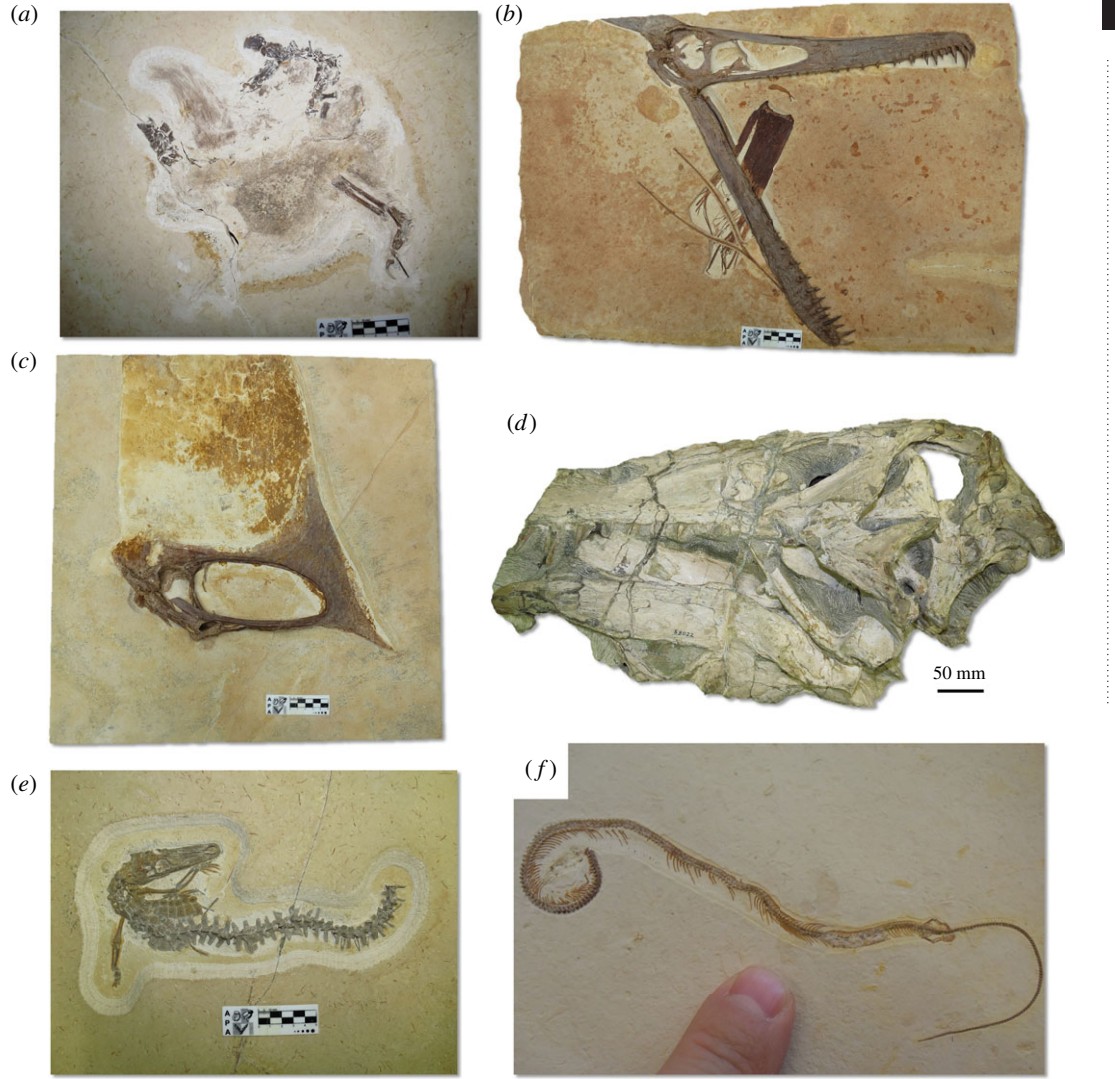

**Figure 2.** Holotype vertebrate fossils from Araripe Basin, Brazil, stored in foreign collections. (*a*) SMNK PAL 29241, proposed holotype skeleton of the feathered dinosaur '*Ubirajara jubatus*' [26], (publication retracted by publisher), (*b*) SMNK PAL 3828, holotype of the pterosaur *Ludodactylus sibbicki* [28], (*c*) SMNK 2344 PAL holotype of the pterosaur *Tupandactylus navigans* [95], (*d*) SMNS 58022 holotype of the dinosaur *Irritator challengeri* [30] (*e*) SMNK PAL 3804, holotype of the crocodyliform *Susisuchus anatoceps* [96], (*f*) private collection BMMS BK 2-2, holotype of the putative legged-snake *Tetrapodophis amplectus* [27], currently interpreted as an aquatic lizard [33], photograph by Michael Caldwell. Abbreviations: SMNK, State Museum of Natural History Karlsruhe, Germany; SMNS, State Museum of Natural History Stuttgart.

in Brazil and Mexico (box 1). We chose this time period as (i) this is when the most research has been carried out in the two basins that we chose as case studies and (ii) many of the relevant laws and regulations were established just before or in 1990. We also propose alternatives and recommendations to scientists, journals, research institutions and government agencies in order to overcome these practices and ensure that future palaeontological research is more ethical and sustainable.

# 2. Legal framework

## 2.1. Mexico

A law regarding archaeological monuments was established in Mexico in 1897 [43], as a reaction to the looting of the Mayan site of Chitchén Itzá by Edward Thompson, then the USA consul in Mérida, Yucatán [44,45]. Currently, the law in force is The Federal Law of Archaeological, Artistic and Historic Monuments and Zones, published in 1972 [46]. The Instituto Nacional de Antropología e Historia,

> **Box 1.** Clarification
>
> All views expressed in this paper rely solely on information, or lack thereof, provided in the publications discussed herein. We do not assume that the authors of the publications here discussed have violated or intended to violate any local laws or regulations. Neither do we assume that all of the co-authors of a particular publication concur with irregular or unethical practices eventually made by another co-author or by an institution.

> **Box 2.** Recommendations for palaeontological studies in the country by National Council of Palaeontology (INAH) [53]:
>
> In the event of the intervention of an academic partnership from foreign institutions as co-responsible for the project, the Palaeontology Council must be notified in a timely manner of their participation and the work they will carry out within the research project. The co-manager of the foreign institution must deliver in writing and with a handwritten signature an official letter in which they undertake to send INAH a report of the results obtained from their participation, as well as the products generated, once the project is concluded.

INAH (National Institute of Anthropology and History), was created through an organic law in 1939 [44,47] in order to protect this heritage. A presidential decree issued in 1986 [48], added the article 28bis to the law from 1972, and reformed the Organic Law of INAH, making it responsible for overseeing any activities involving the discovery and treatment of any fossil material, the delimitation of the boundaries of a fossiliferous site, and the safeguarding of the material in a collection [17,49]. The law declares that fossils are property of the Mexican Federation even if they are under custody of a private person (Articles 27, 28 and 28bis) [46]. Private collections, i.e. fossil collections owned by private individuals or companies, must be registered by INAH [50]. Fossils in private collections are inalienable and imprescriptible, i.e. once registered, they cannot be transferred to other collections [46]. Since 1986, Mexican law explicitly forbids fossils to be commercially traded in Mexico [48,51]. In 1994, INAH created the National Council of Palaeontology to form a multidisciplinary and inter-institutional group with the aim of reaching an agreement on what, how and why to legislate the research on the palaeontological heritage in Mexico [22,52]. Mexican law does not formally require foreign palaeontologists and institutions to work with a local partner. However, the National Council of Palaeontology recently released a series of recommendations for palaeontological studies in the country [53] advising foreign parties that wish to work in the country to notify the council beforehand (box 2). The original Mexican laws and their English translations are available in the electronic supplementary material, appendix B.

## 2.2. Brazil

In Brazil, fossils are protected by Decree 4.146, published in 1942 [54], which states that fossils cannot be privately owned as they belong to the Union and that fossil collecting requires authorization from Agência Nacional de Mineração, ANM (National Mining Agency, formerly the National Department of Mineral Production). In 1990, the Brazilian government published Decree 98.830 (box 3) [18] to regulate foreign scientific expeditions that collect biological or palaeontological material (i.e. fossils) in the country. This law is regulated by Ordinance 55 from Ministério de Ciência Tecnologia e Inovação, MCTI, formerly MCT [55] (see box 3). According to this legislation [55], any foreign party that wishes to permanently export specimens from Brazil must have a permit from MCTI and a partnership with a Brazilian scientific institution (who will be in charge of applying for the permit). Furthermore, Decree 98.830 explicitly states that fossil holotypes, name-bearing specimens that make up 30% of any collected taxon, and other specimens 'whose permanence in the country is of national interest' cannot be exported. A recent ordinance issued by ANM [56] reinforces the necessity for foreign palaeontologists to comply with requirements stipulated by MCTI's ordinance of 1990. The original Brazilian laws and their English translations are available in the electronic supplementary material, appendix A.

> **Box 3.** Portions of Decree 98.830 from 1990 and Ordinance 55 from 14 March 1990 of the Minister of Science and Technology that are relevant to foreign palaeontologists.
>
> Decree 98.830 from 1990
>
> Article 3. The activities referred to in Article l will only be authorized as long as there are co-participation and co-responsibility of a Brazilian institution with a recognized technical-scientific concept in the research field correlated with the work to be developed, according to the assessment of the National Council for Scientific and Technological Development (CNPq).
>
> Ordinance 55 from 14 March 1990 of the Minister of Science and Technology: 'Regulates the collection of scientific material by foreigners, according to Decree 98.830/1990'.
>
> 42 – The MCT, through the co-participant and co-responsible Brazilian institution, will retain, from the collected material, for the destination to Brazilian scientific institutions, the following items:
>
> (…)
>
> (e) all fossil type-specimens;
>
> (f) at least 30% of the specimens of each taxon identified at any time;
>
> (g) other specimens, data or materials, whose permanence in the country is of national interest.

## 2.3. UNESCO convention on illicit trafficking

The UNESCO 1970 Convention on the 'Means of Prohibiting and Preventing the Illicit Import, Export and Transfer of Ownership of Cultural Property' [57] was signed in November 1970 and came into effect in April 1972 as a response to the growth in the illicit market of cultural property since the 1950s. The 1970 Convention promotes international cooperation between countries as a means to protect cultural heritage and is theoretically central to preventing the illicit trafficking of cultural property. Signatories of the convention acknowledge that the 'illicit import, export and transfer of ownership of cultural property is one of the main causes of the impoverishment of the cultural heritage of the countries of origin of such property' and as such, the import, export and transfer of ownership is prohibited under the convention. Thus, the signatories of the convention are required to enact national laws and services for the protection of the cultural heritage. They are also expected to take the appropriate measures so that museums and other institutions within their territories are prevented from acquiring illegally exported cultural property from another country, as well as to cooperate with restitution of the object(s). This convention was ratified by several nation states including Brazil (in 1973) and Mexico (in 1972), both of whom consider palaeontological objects or sites as part of the cultural property, as does Germany (in 2007). Japan (in 2002) includes geological features among other things that are protected by the cultural property law.

## 3. Methods

Our survey is divided into two study cases: Northeastern Mexico (Sabinas, La Popa and Parras basins) and Northeastern Brazil (Araripe Basin). For the first study case of Northeastern Mexico, we compiled a list of publications from the last three decades (1990–2021) on Jurassic and Cretaceous macrofossils of these Mexican basins (minus plants), and other sites of that age in the states of Coahuila and Nuevo León. As a starting point, we looked for publications in which the lead or corresponding author is affiliated to a foreign institution (i.e. not from Mexico), except students (we considered them according to their nationality not their affiliation). We searched for publications in English using the web search engine Google Scholar. Keywords used: Sabinas, La Popa, Parras, vertebrate, invertebrate, fossil, Mexico, Cretaceous, Jurassic (in both English and Spanish). The keywords were used in different combinations (e.g. La Popa + Cretaceous + Invertebrate; Sabinas + vertebrate + fossil) and the top 100 results (pages 1–10) were considered. To these results, we added our previous knowledge of what has been published in our fields of study, in order to provide a list as complete as possible. For example, if a publication was not retrieved during searches, but it was known to the authors, it was included. This added a few publications from smaller journals that are not indexed by Google Scholar or did not show up in the first 100 search results. When an author was found to have multiple publications on the topic, we searched for their Google Scholar or ResearchGate profile to locate more related

publications. Publications that were not focused on fossil localities from these basins were ignored (for example, taxonomic revisions). A preliminary search found no publications led by foreigners on Jurassic or Cretaceous plants in the region, for that reason we included only vertebrates and invertebrates in the list of publications. Publications led by local authors were also included in the tables for comparative purposes. We then checked for a series of factors that may characterize colonial science practices: (i) no local authors; (ii) the local author is not affiliated to a research institution; (iii) absence of collecting permits (if applicable); (iv) fossil stored in private collection; (v) fossil not returned to the country of origin; (vi) no mention of export permits (if applicable); (vii) probable purchase of fossils. The results are presented in electronic supplementary material, table S1.

For the second case study, the Brazilian Araripe Basin, we compiled a list of scientific publications from the last three decades (1990–2020) on Cretaceous macrofossils. Due to the large number of publications on Araripe fossils, we only considered those describing new genera or species (i.e. specimens designated as holotypes) for this basin, and we restricted the search to vertebrates and plants (but see below). We looked for publications where the lead or corresponding author was affiliated with a foreign institution (i.e. not based in Brazil). Like the previous search, this search was performed in Google Scholar, with combinations of the keywords: Araripe, Crato, vertebrate, reptile, fish, plants, Cretaceous (English only). In addition, we consulted a review on Araripe fossils to identify publications that could have been omitted from our searches [58]. As in the survey for the previous case study, we also looked at author profiles on Google Scholar and ResearchGate to attempt locating more publications on the topic. Publications led by local authors were also included in the tables for comparative purposes. We then checked for a series of factors that may characterize colonial science practices, see (i)–(vii) as outlined above. The list of publications is available as electronic supplementary material, table S2.

Brazilian palaeontologists associated with a state or federal research institution are not required to obtain a permit to collect fossils; they must only communicate to Agência Nacional de Mineração (ANM, formerly DNPM) that fieldwork will take place. For foreign authors, we only considered the fieldwork permit to be applicable when the authors explicitly report that the fossil in the publication was collected by them (which is rarely the case). The purchase of fossils was considered as a valid possibility in cases where: the fossils are kept in foreign institutions; exportation permits are not mentioned; fieldwork is mentioned, but it is unclear if the authors collected the fossil themselves; provenance data is vague and/or fieldwork is not mentioned at all (i.e. the fossil suddenly 'appears' in a foreign collection); or purchase is directly acknowledged in the publications.

A preliminary list of publications on Araripe arthropod new genera or species reposited in foreign collections (i.e. outside Brazil) was also compiled (electronic supplementary material, table S3). The list includes publications ranging from 1990 to April 2021. Google Scholar search engine was used with combinations of the keywords: Araripe, fossil, Odonata, Ephemeroptera, Orthoptera, Isoptera, Dermaptera, Hemiptera, Coleoptera, Hymenoptera, Neuroptera, Arachnida, Chilopoda, Crustacea (English only). A review of fossil insects was also consulted [59]. The only purpose of this table was to further help estimate the number of fossil holotypes that were exported from Brazil in the last decades. Due to the unfinished nature of this list, these publications were not used in the statistics presented in our study.

Finally, a Twitter analysis on the repercussions of the 'U. jubatus' case was also carried out. Tweets with the hashtag #UbirajarabelongstoBR were collected using the Twitter Standard Search API [60], from 10 December 2020 to 1 April 2021. A graphic was plotted using a total of 2908 original tweets (including quote-retweets), 10 721 retweets, and 61 073 likes found in that time frame.

# 4. Case studies

## 4.1. Case study 1: palaeontology in the Sabinas, La Popa and Parras basins of Mexico

In northeastern Mexico there are several fossil sites with great palaeontological interest: Parras, La Popa, Rincón Colorado, Múzquiz and Vallecillo (figure 3a,b). They are included within the Sabinas, Parras and La Popa basins, that extend from the centre of Coahuila to the southeast of Nuevo León [61]. Most of these sites are popularly known for the presence of dinosaur fossils, but also contain fossils of invertebrates, plants and several aquatic vertebrates (figure 1). The Muzquiz and Vallecillo deposits support the economic activities of the local community, where flagstone material from these deposits is commercially exploited for construction. During the extraction of the slabs, fossils of exceptional

**8**

**Figure 3.** Fossil sites at the Sabinas, La Popa and Parras basins (NE Mexico) and Araripe Basin (NE Brazil). (*a*) La Mula Quarry, North of Múzquiz, Coahuila. (*b*) Vallecillo Quarry in Nuevo León State, with quarry worker Ramón Ramírez. (*c*) Nova Olinda Quarry in Ceará State. Photographs (*a*) courtesy of Alberto Blanco-Piñón, and (*b*) by Selene Velázquez.

preservation can often be found. Some may be destroyed in the process of extracting flagstone slabs, others are kept by quarry workers [62], some others end up on the illicit market.

One of the most important sites considered in this study is the Vallecillo quarry in Nuevo León, which was exploited for its lead, zinc and silver outcrops. Currently, construction materials are extracted from the quarry. In the 1980s, inhabitants of the Vallecillo village started to find several well-preserved vertebrate and invertebrate fossils [63]. The palaeontological value of the locality was recognized as early as 1997 [64], but research on fossils from this locality has been intensively carried out by foreign researchers. Material from the quarry was first exported in 1999–2000 to the University of Karlsruhe in Germany, where it remained until it was returned to Nuevo León in 2007 and deposited in the Museo La Plomada [65].

We found 128 publications on Jurassic–Cretaceous macrofossils (excluding plants) from Sabinas, La Popa and Parras basins, and other sites in Coahuila and Nuevo León states in NE Mexico, published between 1990–2021 (see electronic supplementary material, table S1). Foreign researchers led 46.88% of the publications (figure 4). Most of this foreign-led research (51.67%) does not include local authors (i.e. an author based at a Mexican scientific/academic institution according to the affiliations of the publications). In four publications [32,66–68], the local authors included within the author list are not affiliated with any scientific institution. Five publications are based on fossils that are stored in private collections (i.e. not in a recognized museum or other official scientific repository), two of which describe new genera and species (holotypes) [25,32]. Finally, two mention the purchase of a fossil [32,39] whereas another mentions that a specimen 'was obtained by a private collector from a local quarry worker' [69]. We found no record of Mexican fossils being stored in foreign collections, except for a study published in 1990 [70] on a collection of fossils claimed to have been collected in the 1930s.

Some recent studies of fossils from this area warrant a closer look. Frey *et al.* [25] describe the plesiosaur *M. fernandezi* (figure 1*e*) from Vallecillo, in the Sabinas Basin, declaring it will be stored

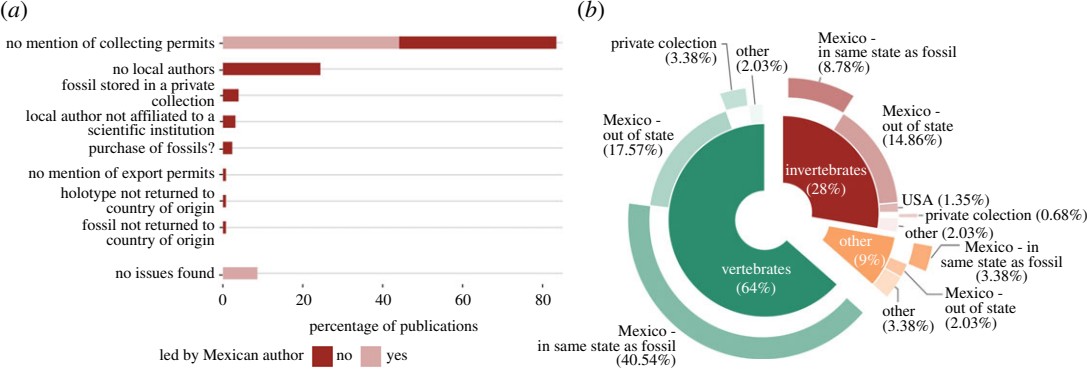

**Figure 4.** Publications on Jurassic and Cretaceous fossils from Sabinas, La Popa and Parras basins, and other sites in Coahuila and Nuevo León states between 1990–2021 (plants and microfossils excluded). (*a*) Issues detected in the publications. (*b*) Current location of the fossils. See electronic supplementary material, table S1, for list of publications.

and accessible at Museo del Desierto in Saltillo, Coahuila, but only three pages later the authors contradict themselves and state that this plesiosaur specimen is reposited at Museo Papalote Verde in Monterrey (Nuevo León). The latter is a museum aimed at children, exhibiting fossil replicas, and does not have an INAH registration nor a permit to store fossils. The *M. fernandezi* specimen was published using the INAH register number CPC RFG 2544 P.F.1 (*sic*) which refers to a private collection, despite the claims made in the paper. INAH uses a standard protocol for registration numbers, in which the suffix 'PF' (=*persona física*) refers to a private individual [71]. The prefix 'CPC RFG' is not used by INAH, and probably was confused with a registration code from Museo del Desierto, which actually uses 'CPC' (=Colección Paleontológica de Coahuila) in its catalogue, and a misspelling of REG (=*registro*). According to the INAH standard code, the correct registration number of this fossil should be REG2544PF1, i.e. the collection of Mauricio Fernández (REG2544PF), fossil number 1. As a fossil registered to a private individual by INAH, the only known specimen of the plesiosaur *M. fernandezi* cannot be transferred to a museum as per Federal Law on Archaeological, Artistic and Historical Zones and Monuments [46]. Furthermore, this work was published at Boletín de la Sociedad Geológica Mexicana, a journal that, unfortunately, does not formally require fossils to be accessible in a public collection or recognized research institution [72].

The chondrichthyan *A. milarcae* (figure 1*f*) also found at the Vallecillo quarry and recently described by Vullo *et al.* [32] in the journal *Science*—which has general editorial policies with regards to ethics—has already stirred controversy [73]. The only local person involved in this study is neither affiliated with an institution nor is he a scientist, as he admits in an interview [74] despite being listed as an 'independent researcher'. Vullo *et al.* [32] openly recognized that the fossil was purchased from a quarry (see original version of Supplementary Materials [75]). The holotype specimen was reported to be registered as INAH 2544 P.F.17 (*sic*). Again, this is an incorrect catalogue number *sensu* INAH standards. The correct INAH register would be REG2544PF17, i.e. private collection of Mauricio Fernández (REG2544PF) fossil number 17. In a statement, Vullo *et al.* [75] promised that the specimen would be available to researchers in a museum that had in fact not yet been built at the time of publication. As mentioned above, fossils registered in a private collection in Mexico cannot be transferred to other collections [46], hence, this specimen will remain under the custody of the private collector even if it is loaned to a museum. A correction to the Supplementary Materials was published by Vullo *et al.* on 8 April 2021 [76] and an erratum only one week later [77]. The erratum states that the fossil will be housed at Museo del Desierto in Saltillo, Coahuila, until the new museum opens. In both amended versions of the supplement [76,77], Vullo *et al.* restated the registration number incorrectly as INAH 2544 P.F.17 (*sic*), and did not mention that this is a private collection [71], nor that the fossil was purchased, as originally stated by the authors. Moreover, the owner of the *A. milarcae* specimen recently declared that he bought the rock that contains the fossil [73].

A general issue, for both foreign and local parties, is the absence of collecting permits being reported in publications, despite these permits being a lawful requirement for work at fossil sites in Mexico [46] (figure 4). When a palaeontological research project is registered through the Council of Palaeontology of INAH, the application for a collecting permit is included. Permits are then generated if the project is approved. Collecting permits may be requested by INAH personnel or local authorities during fieldwork on Federal areas, but this rarely occurs. Traditionally, INAH has advised palaeontologists to

register their projects when they involve the temporary export of specimens or when they are carried out by foreign researchers only. Recently, the Council of Palaeontology at INAH has published new guidelines encouraging all researchers to register their palaeontological projects [53]. It is worth mentioning that some journals that publish palaeontological studies only recently began requesting permit information to be disclosed, while the majority do not yet formally request this information at all.

Some of the issues described above represent clear examples of scientific colonialism. It is remarkable that despite the presence of local expertise, which was responsible for the majority of the research output in the same period ($n = 68$), most studies led by foreign palaeontologists do not include local Mexican researchers. It is also questionable that some of these studies include local, independent authors but not local institutions, which may suggest a general aim of non-cooperation with local scientists. In addition, five of these studies were based on fossils from private collections, which casts doubts on access and reproducibility of the results (see below). By contrast, none of the studies led by Mexican authors used fossils from private collections.

## 4.2. Case study 2: palaeontology in the Araripe Basin of Brazil

The Araripe Basin is located in the northeast of Brazil (figure 3c), a region with the highest concentration of poverty in the country (47.9% of the population in the region) [78]. Fossiliferous outcrops occur in the south of the Ceará State, northwest of the Pernambuco State and east of Piauí State, encompassing several municipalities with low Human Development Index (HDI; a composite index of life expectancy, education and per capita income indicators) [79]. The Araripe Basin has long been a source for the illicit fossil market [42,80,81]. The earliest recognition of palaeontological specimens from the Araripe Basin was in an 1800 letter from the Brazilian naturalist João da Silva Feijó, and the reported fossils were promptly sent to the Portuguese Academy of Sciences, in Lisbon, where they still reside to this day [82]. The first formal account of an Araripe fossil was an illustration in the famous book series *Reise im Brasilien* (1823–1831) by two German naturalists who were members of the entourage of Maria Leopoldina of Austria, shortly before she became Empress consort of Brazil [83]. After the Brazilian expedition of George Gardner (1836–1841), who sent abundant Brazilian fossil fishes to the ichthyologist and proponent of scientific racism Louis Agassiz [84] in the USA, the importance of Araripe specimens became clear overseas, stimulating the exploitation, trade and assembly of huge collections. Some of these collections were fortunately destined to publicly accessible institutions, such as the Axelrod Fossil Fish Collection (donated to the American Museum of Natural History, New York) and the Desirée Collection (donated to the Museu Nacional, Rio de Janeiro). It is, however, impossible to estimate the number of private collections containing Brazilian fossils, and the (illegal) trade of these specimens persists to this day [85–88].

We found 71 publications on Cretaceous macrofossils (excluding invertebrates and non-holotype material) from the Araripe Basin, published between 1990 and 2020 (see electronic supplementary material, table S2 for list of publications). The majority of these publications (59.15%), were led by foreign researchers, and over half of foreign-led publications (57.14%) showed no evidence of collaboration with local Brazilian researchers (figure 5). A large proportion (88%) of the fossils described in these publications (all holotype specimens) were taken from Brazil to be housed in foreign museum collections and have not yet been returned. Among the publications describing fossils that were permanently taken to foreign collections, none reported export permits. Only one [89] reported that the specimens were collected during fieldwork conducted by the authors, yet did not mention a collecting permit as required by law. Several publications provide only vague statements of provenance (e.g. 'Araripe Basin, Brazil' [90], 'Araripe Plateau, Brazil' [91]), and do not mention fieldwork nor explain how the fossils ended in foreign collections. This fact, together with the absence of reported exportation permits, leads us to consider that these fossils may have been purchased (figure 5). Some publications state that the fossils were 'obtained from a quarry workman' [92] or 'from a fossil digger' [93] and eight publications [28,30,94–99] directly acknowledge that the specimen was purchased.

Brazilian researchers, on the other hand, are responsible for 40.85% ($n = 29$) of the published research on new species of Araripe vertebrates and plants during the same period (figure 5). Three of these publications are based on fossils reposited in foreign collections, with no mention of export permits; one of these fossils was supposedly collected in the 1960s [100], whereas the two others [101,102] lack any data on provenance and probably were purchased by the museums.

Two recent publications on particularly high-profile fossils deserve our attention here due to the questionable practices exposed therein. The putative legged-snake *T. amplectus* from Araripe caused

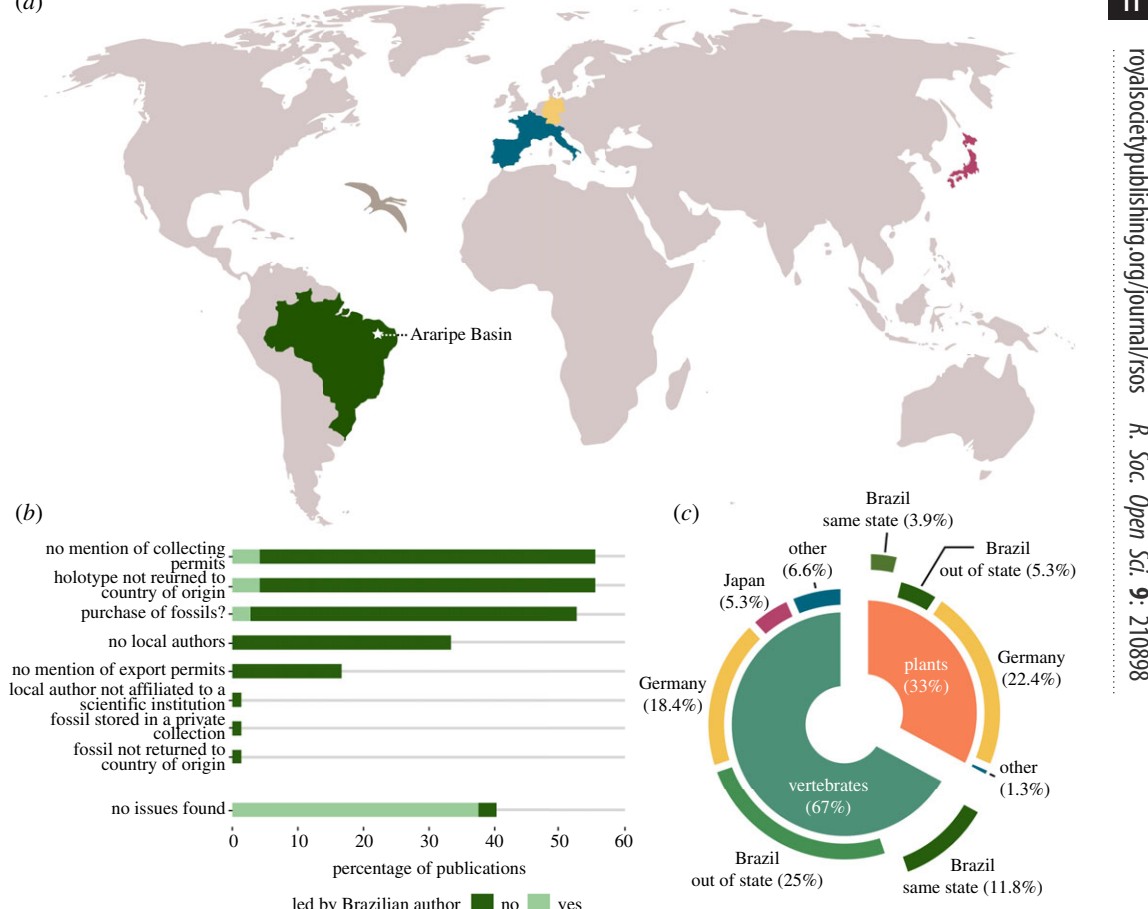

(a)

Araripe Basin

(b)

no mention of collecting permits
holotype not reurned to country of origin
purchase of fossils?
no local authors
no mention of export permits
local author not affiliated to a scientific institution
fossil stored in a private collection
fossil not returned to country of origin

no issues found

percentage of publications

led by Brazilian author    no    yes

(c)

Brazil same state (3.9%)

other (6.6%)

Japan (5.3%)

Brazil out of state (5.3%)

Germany (22.4%)

plants (33%)

Germany (18.4%)

other (1.3%)

vertebrates (67%)

Brazil out of state (25%)

Brazil same state (11.8%)

**Figure 5.** Publications on Cretaceous fossils from Araripe Basin, Brazil, between 1990 and 2020 (only holotypes, invertebrates excluded). See electronic supplementary material, table S2, for list of publications and description of methods used. (a,c) Current location of the fossils. (b) Issues detected in the publications.

considerable controversy [103,104] when it was published in 2015 in the journal *Science* by Martill *et al.* [27] (figure 2f). To start with, the publication did not involve any Brazilian researchers or institutions. The authors also claimed that the specimen was permanently available in a museum. The fossil, however, belongs to a private collector in Germany [103,105,106] and access to this specimen has reportedly been made difficult for other researchers who wish to study it [103,105]. Finally, the authors did not provide any evidence that the fossil was legally collected and exported from Brazil.

More recently in 2020, a publication describing the dinosaur '*U. jubatus*' (figure 2a), also from the Araripe Basin, did not include Brazilian researchers or institutions. The specimen was collected and removed from Brazil after 1990, so in accordance with Brazilian law, the work should have been conducted through a partnership with a Brazilian institution, and the specimen should have been accompanied by a permit from MCTI. The *in press* article appeared in the journal *Cretaceous Research*, but was temporarily removed two weeks later by the editor [26], pending an investigation. This attracted considerable media attention. Through a journalistic report [107], E. Frey, co-author of the publication and curator at the State Museum of Natural History Karlsruhe (SMNK), Germany, where the fossil is currently housed, presented a document from 1995 [108] signed by José Betimar Melo Filgueira, an agent at the regional office of the National Department of Mineral Production. This document is surprisingly vague, and does not specify how many or which kind of fossils were exported, just 'two boxes with limestone samples containing fossils'. There was also no information on whether this was a temporary or permanent exportation, nor mention of any collaborating Brazilian institution. The aforementioned agent has previously collaborated with one of the authors [109] of the '*U. jubatus*' paper around the same time that the specimen was exported from Brazil, which may represent a conflict of interest. Furthermore, no mention is made by the authors of the study of the necessary export authorization from MCTI, legally required as per Decree 98.830 [18].

It was recently revealed that, contradicting Frey's version, the specimen of 'U. jubatus' was not exported in 1995 nor transported to Germany by this individual but actually acquired by a private company in 2006 and sold to SMNK in 2009 [110]. The absence of satisfactory proof regarding the legal status of the specimen prompted the journal Cretaceous Research to permanently remove the 'U. jubatus' paper in September 2021.

Several of these studies led by foreign researchers represent clear cases of scientific colonialism. They also display a disregard for local expertise to an even greater extent than observed in the examples from Mexico. To make matters worse, the overwhelming majority of these studies are based on fossils likely to be both illegally purchased and exported. Our survey shows that the primary destinations of most of the illicitly exported fossils are the Museum für Naturkunde Berlin (13 plant holotypes) and Staatliches Museum für Naturkunde Karlsruhe (10 vertebrate holotypes) (see electronic supplementary material, table S2). Our study was limited to fossil holotypes, and we did not include arthropods and other invertebrates, therefore the above numbers probably represent only a small portion of the fossils that have been irregularly taken from Brazil to foreign institutions. Our preliminary search for Araripe arthropod publications on Google Scholar (see electronic supplementary material, table S3), however, indicates that numerous holotypes of insects, arachnids and chilopods have been illegally transferred to German collections, including at least 47 to Staatliches Museum für Naturkunde Stuttgart, seven to Senckenberg Museum in Frankfurt am Main, four to Museum für Naturkunde Berlin, four to Staatliches Museum für Naturkunde Karlsruhe, and several others to various other institutions and private collections, totaling a minimum of 90 Araripe holotypes. In addition, our survey of publications located at least four vertebrate holotypes and seven arthropod holotypes housed in Japanese collections.

# 5. Beyond the Sabinas and Araripe basins

Colonial palaeontology practices in Brazil and Mexico are not limited to Sabinas, La Popa, Parras and Araripe basins. In Mexico, important Pleistocene mammal specimens from the Yucatán Peninsula have been targeted [111–116], as have the fossil deposits from the Miocene strata of the Acre Basin in Brazil [117,118]. A 2012 study found that foreign-led research has extensively been conducted in several northern Mexican states (Baja California Sur, Coahuila and Nuevo León) [119]. Furthermore, it identified research carried out exclusively by US-based researchers in three states: Guanajuato, Jalisco and Sonora [119]. The petrified wood from the Permian Pedra de Fogo Formation in Brazil has also been the subject of these same practices [120–123] with a number of holotypes irregularly stored at the Chemnitz Museum in Germany.

Latin America is not the only region affected by such practices. Other countries such as Morocco, Mongolia and Myanmar (Burma) have also been on the receiving end of palaeontological colonialism. Moroccan palaeontology has been largely fed by commercial fossil exports to mainly Europe and North America despite the fact that Morocco is a signatory of the UNESCO 1970 Convention and has enacted a 1994 ministerial order explicitly prohibiting the export of fossils [124,125]. While exporters usually obtain a large income from fossil exports (estimated at US$100 000 per year), these fossils are originally found and excavated by local people living in marginalized areas who derive very limited income from these specimens [125]. As a result, Morocco has been the source of a great number of exceptional specimens without any contextual information on the geography or geology, a large number of them being described and published by a commercial collector lacking a scientific background, and/or without any participation from Moroccan research institutions [126]. In 2019, the Moroccan government drafted a decree with specific recommendations regarding authorized exports and loans—a step forward regarding the protection of palaeontological heritage as well as the livelihoods of those dependent on it [125].

Similarly, palaeontological research in the Mongolian Gobi desert had been occurring for almost a century, but until recently, this palaeontological heritage was largely unknown to the general public in Mongolia [127]. This has changed in recent years due to efforts by the Mongolian government assisted by other parties in other countries to arrange the repatriation of fossils (See §9 below) as well as an international initiative for scientific outreach largely driven by the non-profit Institute for the Study of Mongolian Dinosaurs [128]. Lastly, the realization by the palaeontological community of the controversial provenance of Myanmar (Burmese) amber and its link to smuggling and human rights violations has sparked reforms across the palaeontological community [129]. The reaction of some professional societies and journals was to declare a moratorium on amber material from Myanmar

[130–132]. This set a precedent for both professional societies and journals to adapt their guidelines and codes of conduct to address and curtail unjust and unethical research practices, although some journals and societies have resisted adopting these standards.

# 6. In defence of scientific colonialism in palaeontology and beyond

There are several arguments commonly used by academic researchers and fossil collectors to defend unethical scientific practices which result in scientific colonialism, not only in Brazil and Mexico but also in other countries.

## 6.1. Fossils should be considered global heritage, not national heritage [23,24,133]

This is one of the most popular justifications for colonial practices in palaeontology. The argument is that unlike archaeological artefacts, fossils are not related to any geopolitical boundaries, history or culture of a specific people, region or country [23,24,133]. This is an incorrect assumption as fossils have been known to humankind since ancient times and, in several documented cases, have become part of the local folklore and myth [134–137]. Furthermore, even with archaeological artefacts and heritage sites, connection between the people who produced the monuments or artefacts and modern nation states is often tenuous. For example, the claim related to geopolitics could also be applied to other sites or objects, such as Stonehenge and Sutton Hoo in England, Lascaux in France or the Colosseum in Italy. These sites existed long before modern nation states were established and in several cases were produced by people without a clear connection to modern residents of the nation state, but saying that these are not to be considered the national heritage of those countries respectively would be broadly rejected as ridiculous. Furthermore, the 1970 UNESCO Convention, the basis for both national and international legislations, states that the definition of such objects belongs to the nation state in which they are located [138]. We argue that fossils add to local heritage in the form of scientific and even historic and cultural importance or value. The idea that natural history should be considered 'global heritage' stems from the systems that aided the construction of the colonies as 'living laboratories', where research practices and experiments that would not have been feasible in the colonizing countries would have been carried out without limitations [3]. In the modern world, this line of thinking would lead to a form of neocolonial pillage of the palaeontological resources in countries that are unable to protect them [138]—an archetype of scientific colonialism. This would not only be destructive to the local scientific community, but also to individuals that make use of these fossils for cultural purposes [3,139]. The inadequate, or complete absence of, law enforcement aimed at protecting palaeontological resources has allowed nations with considerably greater access to funding to exploit these resources, hindering scientific development in the country of origin and leading to the skewed pattern of global knowledge production in palaeontology we observe today [14]. It is also important to note that nations with greater access to funding owe this wealth, in great part, to the extractive colonial practices they have been carrying out for centuries. Furthermore, this argument is often, tellingly, not applied to resource-rich Global North countries like the USA, Canada and Italy where local legislation requires compliance with a permitting system in order to make collections (see below), whether this system applies to a subset of public lands (as in the USA) or states (Canada and Italy).

## 6.2. Host countries do not have adequate facilities or personnel to store fossil objects so they are safer in 'Western' museums [140–142]

The case of the National Museum in Rio de Janeiro burning down in 2018 [143] has more recently been used as an example of how fossils are supposedly not 'safe' in Brazil. Targeting only museums in the Global South using this argument is fundamentally wrong. Western museums and historical sites are similarly at risk of being damaged or destroyed, e.g. through fire [144–147], substandard conditions of museums [143,148] and high risk to extreme weather conditions [149–152]. The holotype of the dinosaur *Spinosaurus aegyptiacus*, collected in Egypt and stored in Munich [153], and that of crocodilian *Gryposuchus jessei*, collected in Brazil and stored in Hamburg [154], were both destroyed in Allied bombing campaigns during WWII, together with several other important fossils in museums across Germany [155].

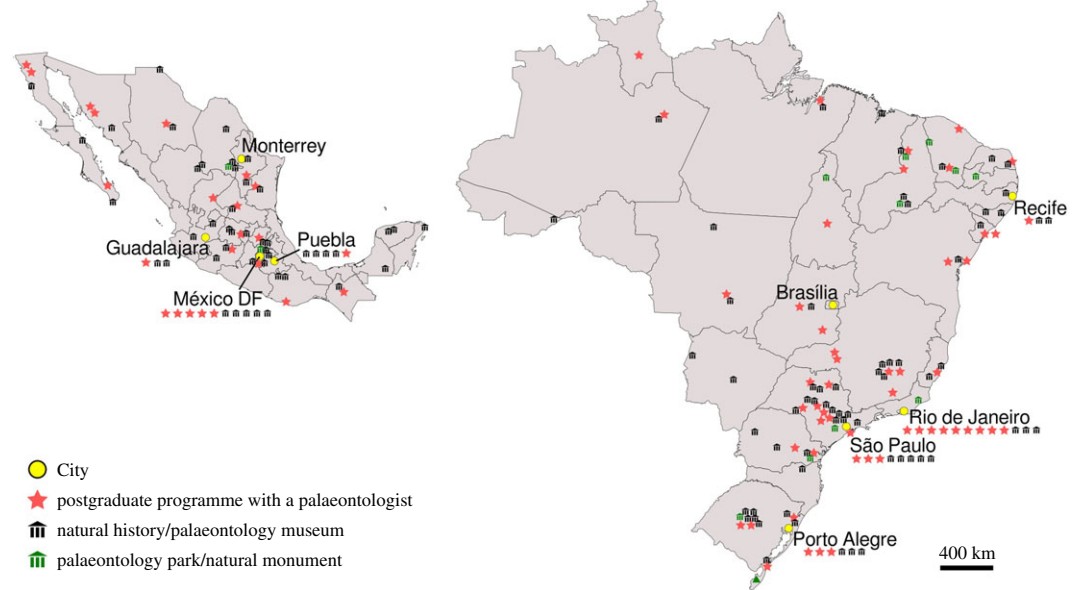

**Figure 6.** Museums, natural monuments and institutions providing postgraduate courses related to palaeontology in Mexico (left) and Brazil (right). See detailed list in electronic supplementary material, tables S4 and S5.

There are several natural history museums in both Brazil and Mexico, some of which are located in the states where the fossils that we mention in this study were collected (figure 6). The Palaeontology Museum 'Plácido Cidade Nuvens' at Santana do Cariri, which contains thousands of fossils from the Araripe Basin, was founded in 1985—presumably before most of the Araripe Basin fossils included in this study were collected. Currently, local researchers associated with this museum coordinate controlled excavations in the region [156–158], and several scientific publications have been produced based on these materials [159–165] (see more publications in electronic supplementary material, table S2). This museum would be the logical place to store a number of holotypes and other important fossils that are now in foreign collections. Furthermore, even if ex-colonies did have insufficient repositories or personnel, this argument ignores the fact that a long history of colonial extraction is the most obvious reason why museum facilities are absent, insufficient or underfunded to begin with.

## 6.3. There is a lack of local scientific expertise, research education and investment in science in lower income countries [166,167]

The number of researchers in high-income countries is generally greater than that in middle- and low-income countries [168]. However, the assumption that no infrastructure for research training nor 'appropriately qualified' local researchers exist in lower income countries (e.g. Brazil or Mexico) is fundamentally flawed. There are many natural history museums and several institutions that offer postgraduate palaeontological courses in both Brazil and Mexico (figure 6). Raja *et al.* [14] also show that Latin American countries contribute significantly to palaeontological research in the region, with Brazil and Mexico being the leading countries in the region after Argentina. At the time of publication, the Brazilian Palaeontological Society (SBP) has 376 associates and the Mexican Palaeontological Society (SOMEXPAL) has 142 members. These numbers should be considered conservative as they may only reflect the population of researchers who choose to have a formal association with these scientific societies. Despite the acknowledged need for even more investment in science in both Brazil and Mexico, the research output from these countries has increased notably in recent years and grows at a faster rate than most Global North counterparts; according to data from the National Science Foundation from the USA, Brazil and Mexico had a growth of 9.13% and 6.8%, respectively, in science publications between 2000 and 2018 (figure 7), with Brazil ranked 11th in world research output. Germany and the UK, on other hand, only increased 2.37% and 1.34%, during the same interval (figure 7). As mentioned above, Mexican researchers were responsible for the majority of publications (51.12%) on Sabinas, La Popa and Parras fossils in the last three decades, whereas Brazilian palaeontologists produced 40.85% of the research on the Araripe Basin in the same

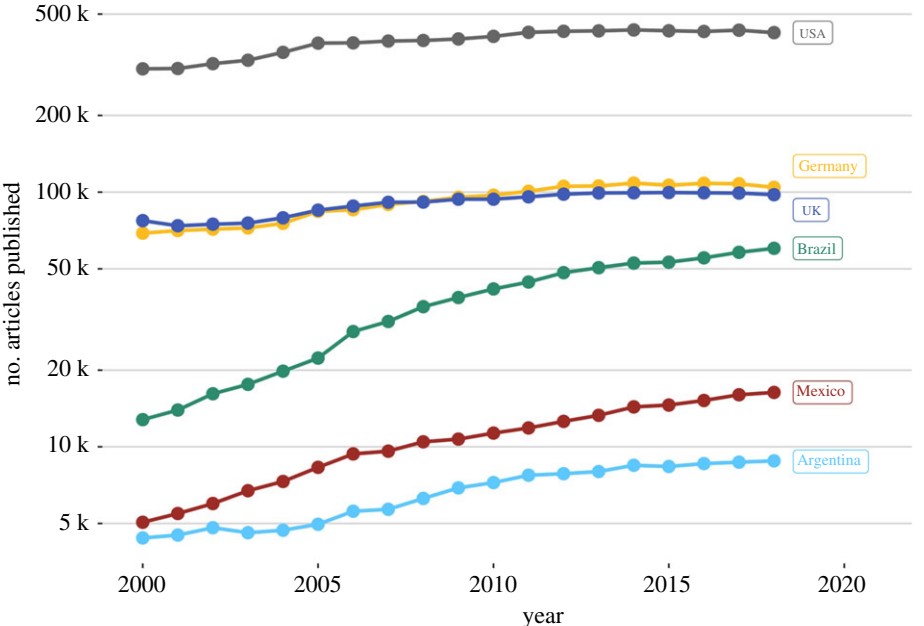

**Figure 7.** Comparison of publications published by country during 2000–2018. Data from National Science Foundation from USA available through The World Bank at: https://data.worldbank.org/indicator/IP.JRN.ARTC.SC.

period. It is also worth mentioning that lower income countries tend to have the highest percentages of open-access publications, thus making their research more widely accessible [169].

## 6.4. There is a disinterest in fossils among the local community [170,171]

Local communities can only express interest in their local heritage when they have adequate access to that heritage. Science outreach activities, including those produced by local museums, not only inform people about their local area but help to generate links between society and the object of dissemination. In addition, they provide information for the public to make sound social, economic and political choices about their own surroundings and resources. Removing fossils from their place of origin results in the deprivation of opportunities to develop an appreciation of and a cultural connection with their palaeontological heritage. Furthermore, publishing any knowledge about fossils from a particular locality in a language other than the local language and without appropriate scientific outreach or public engagement also contributes to depriving the local community of access to that information and leads to further alienation. Several outreach activities that connect palaeontologists, schools, NGOs and the general public currently occur in both Brazil and Mexico (see some examples in figure 8). These activities have had tremendous impacts on the popularization of palaeontology in these countries as well as education, geoconservation and geotourism.

Social media has also played a big role in promoting palaeontology in Brazil and Mexico. Extensive public engagement by Brazilian palaeontologists is precisely the reason why the hashtag #UbirajaraBelongstoBR gained popularity on multiple social media platforms following the controversial publication of the study describing '*U. jubatus*' in December 2020 [172]. The hashtag has been associated with hundreds of thousands of posts on Twitter (figure 9), hundreds of posts on Instagram and Facebook, and at least 150 YouTube videos across 115 different channels. Distinct types of audiences participated in the #UbirajaraBelongstoBR movement: scientific and non-scientific as well as Brazilian and non-Brazilian. Following the movement, dozens of articles in both Portuguese and English were published in Brazilian media [107,173,174], as well as in international news articles [172,175].

## 6.5. Specimens are lost to science if they are not collected and studied [129,176]

This argument is frequently used by palaeontologists to justify poor research practices, even beyond examples of scientific colonialism. Those who use this argument wrongly assume that legislations and guidelines for ethical conduct act as barriers to science. In truth, legislation and guidelines for ethical conduct seek to increase participation in science, particularly from local researchers, and create a more

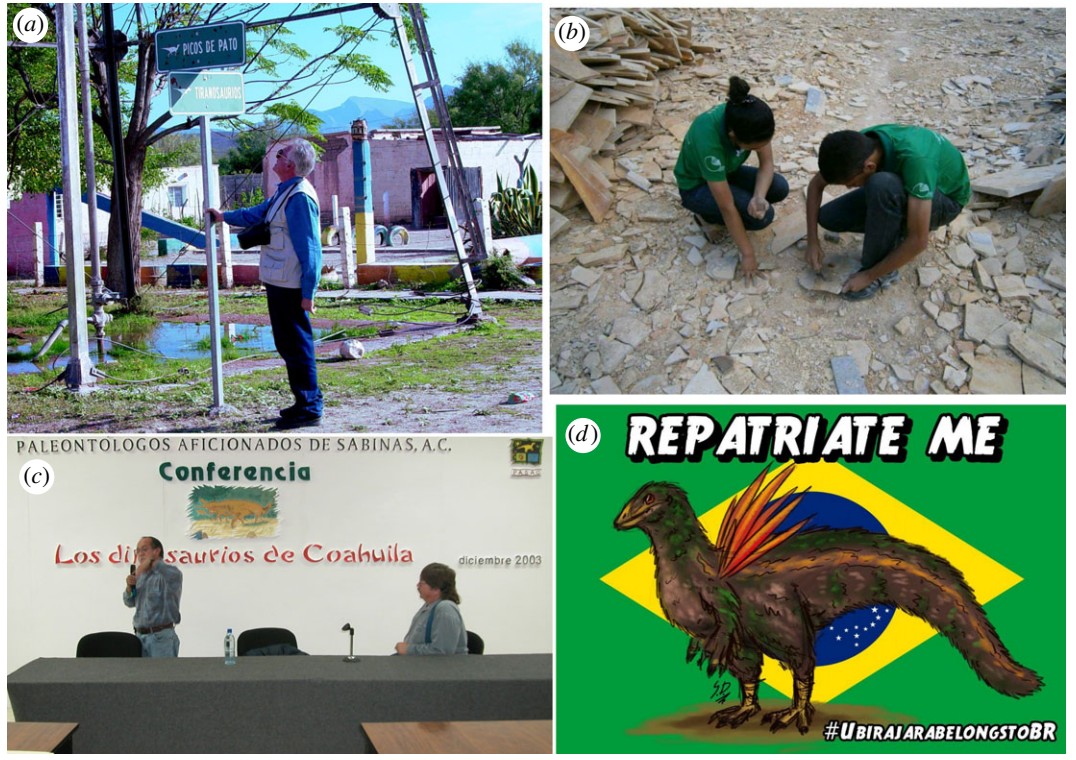

**Figure 8.** Outreach activities and public interest in palaeontology in Brazil and Mexico. (*a*) The cross of Picos de Pato (duck-bill dinosaurs) and Tiranosaurios streets at Rincón Colorado, Coahuila, México (with palaeontologist Giuseppe Leonardi). (*b*) School students learn how to find fossils in the Jovens Paleontólogos (Young Palaeontologists) Project in Nova Olinda, Ceará, Brazil, by Universidade Regional do Cariri (URCA). (*c*) Meeting of Paleontólogos Aficionados de Sabinas A.C. (Civil Association of Amateur Palaeontologists of Sabinas) in Coahuila (René Hernández Rivera and Jim Kirkland seen in the photograph). This association created the Museo Paleontológico de Múzquiz in 2005 [62]. (*d*) Fan art with #UbirajaraBelongstoBR hashtag posted on Twitter in December 2020 (credit: Saulo Daniel Ferreira Pontes, @saulodfp).

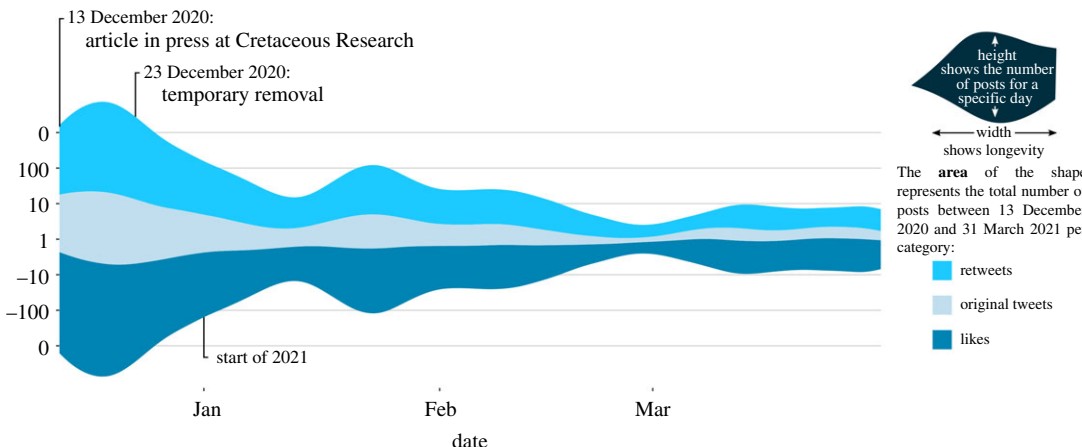

**Figure 9.** Posts on Twitter.com using the hashtag #UbirajaraBelongstoBR between 13 December 2020 and 31 March 2021.

equitable environment for all. Providing that the work complies with the legislation of the country of origin (see below), fossil material can be collected or studied by any researcher with the desire and means to do so. Researchers who use this argument are, therefore, wrongly implying that fossil specimens cannot be adequately collected or studied by local researchers, or that local expertise is absent entirely. This is almost always not the case, as exemplified by the research output by local researchers in Brazil and Mexico (see §6.3 and figure 7).

Fossils are lost every day due to natural and human processes, such as weathering and erosion, natural disasters, quarrying and construction; there is no conceivable way that palaeontologists can

collect and document all fossils that were ever formed. Even when fossils are collected, important contextual data (e.g. stratigraphy, location information, etc.) and even the specimen itself, can become lost through poor collecting procedures, inadequate preservation, disasters, or accidents (see §6.2). This loss of fossil data is not restricted to low-income countries—it occurs worldwide. Arguing that specimens will be lost to science because they have not been collected and studied by 'Western' researchers is deeply rooted in colonialism and is easily exposed by turning to look at fossil specimens from countries in the Global North.

## 6.6. Laws are too complicated or difficult to access [10,11,177]

Although fossil laws in Brazil and Mexico are typically compared with the permissive laws of Germany and the United Kingdom, comparisons are seldom made with countries like Canada or Italy (who strictly regulate collection and export of fossils), the USA (who regulate fossil collection on federal land), or even Australia (who restrict exportation of fossils found in any part of its territory). In Canada, the export of fossil resources is strictly and explicitly regulated at both the federal level by the Canadian Cultural Property Export Control List (C.R.C., c. 448) [178], with further restrictions at the level of individual provinces. Canada also protects several localities of world importance through UNESCO and the national park system. These include the classic Joggins Fossil Cliffs [179] and Dinosaur Provincial Park [180], both UNESCO World Heritage sites, the Burgess Shale, which is protected as part of the Yoho and Kootenay National Parks [181], and the Devonian Escuminac Formation, protected within Parc National de Miguasha [182]. Italy, a high-income European country, also protects its fossil resources under a series of cultural heritage laws, the same legislation that applies to archaeology [183]. Fossil commerce and permanent exportation are illegal, and collecting require an authorization by Soprintendenza Archeologia, Belle Arti e Paesaggio (Superintendence of Archeology, Fine Arts and Landscape) [183]. Italian fossils can be temporarily exported only for study and/or exhibition with the permission of the Soprintendenza. The USA, also, explicitly protects fossils in the public trust, so long as those specimens are collected on public lands [184], and requires that these fossils remain property of the federal government. As public lands represent approximately 25% of the landmass of the USA (including the majority of the fossiliferous mountain west), this represents a substantial restriction on fossil collection and export from the USA. Furthermore, a number of national parks and monuments in the USA have been established in areas of significant palaeontological resources, sometimes explicitly with the intention of preserving those resources. Fossil collecting laws in Australia are variable among its provinces and territories. In Queensland, the Northern Territory and Tasmania, collection of fossils is partly restricted, requiring a licence [185]. The export of fossils from Australia (as well as meteorites) is restricted by the Protection of Movable Cultural Heritage Act 1986 and the associated Regulations 1987, requiring the issue of a permit by an accredited examiner [185]. We are not aware of any criticisms of these heritage laws or assertion that they are unnecessarily nationalist, nor are we aware of major engagement by foreign researchers with groups attempting to circumvent these protections. We, therefore, must conclude that the existence of national heritage laws is not an obstacle to good palaeontological research [186].

However, it can be difficult for foreign researchers to navigate local legislation and bureaucracy, and cooperation with local institutions in Brazil and Mexico is essential in this regard (and required in the case of Brazil). In addition to assisting with legal procedures, the Brazilian or Mexican institution can provide guidance on specific legislation regarding the collection, study and/or temporary export of fossils. The relevant laws that apply to palaeontological work in Brazil and Mexico (translated into English) have been provided in full herein (see electronic supplementary materials, appendices A and B). The bureaucracy that may annoy and frustrate some aids compliance with local legislation so that local heritage is protected.

## 6.7. Commercial exploitation of fossils aids science [187,188]

This assumption suggests that the commercial exploitation of fossil deposits (e.g. mining, quarrying) and the trading of extracted fossils may result in more fossils ultimately being uncovered. However, a greater number of fossils does not necessarily imply a benefit for science. When fossils are openly traded, the exploitation of fossil deposits can become uncontrolled, probably resulting in a loss of important provenance information about that material. The removal of fossils without any documentation of geological information reduces the scientific value of these specimens; new fossil material may lead to the discovery of new species but, without context, these specimens cannot inform on the ecology or

evolution of these organisms. In the case of permissible trade (i.e. allowing only the sale of certain types of fossils and/or fossils from certain locations), as in Morocco, the law of 'supply and demand' must be considered. The commercialization of this type of rare object (i) leads to an increase in inequality in science, by concentrating this type of material in institutions and countries with the financial means to acquire them, (ii) encourages the existence of private collections, which can be an obstacle to the reproducibility of science, and (iii) stimulates the artificial modification of fossils [189,190] to obtain a better market price. The latter can be exemplified by a case from the Araripe Basin itself. Martill *et al*. [30] studied an illicitly acquired, artificially 'enhanced' specimen from Brazil and only discovered these heavy modifications during the course of their work. The case left the authors so 'irritated' that they decided to express this frustration in the name of the new taxon: *Irritator challengeri*.

Commercial exploitation can be an ally in specific cases when it involves controlled, regulated and documented collection of material. Nonetheless, this discussion should be led by the local community, together with effective communication initiatives from palaeontologists and local experts to highlight the scientific and cultural importance of fossils.

# 7. Implications for science and the local community

## 7.1. Private collections can interfere with the reproducibility of science and impede access for both scientists and the general public

Fossil specimens and the data obtained from them need to be accessible by scientists, not only for reproducibility and replicability but also for verification and comparison purposes. A private collection is usually one that is in the ownership of a personal or corporate entity and any access to the material is at the discretion of the owner [191]. However, unless the appropriate arrangements are made, there is a risk of losing these fossil materials due to changing circumstances, e.g. death or illness, or changes in personal finances that necessitate selling off parts of a collection. Current and future research relies on the permanent accessibility and stable storage of these materials, which unless placed in a public trust, is rarely the case for privately owned collections [130,192]. Private collections also hinder access of information to the general public. Science is a public endeavour, often funded by tax money, and as such, scientists have a responsibility to relay the findings and provide access to materials—through museums—to the public. When fossils are stored in private collections, this negates the public's ability to not only enjoy, but also scrutinize scientific research arising from public funds.

## 7.2. The purchase of fossils does not benefit the local community in the long term

Fossil deposits are finite. With the depletion of the resource, the living wage earned from any associated trade quickly becomes inaccessible to people financially dependent on it. In addition, the impacts of mining activity, including environmental damages, will be borne by the community long after the resource is exhausted. It is a trade-off whose long-term cost generally falls on the most vulnerable citizens.

The countries and provinces discussed in this study are economically vulnerable, with mid–low Human Development Indices, and low education levels in contrast to European standards [79,193]. The local communities are thus vulnerable to the exploitation of third parties, and the economic benefit most of the time does not reach the segment of the society that most needs it [78] (e.g. see a survey of Araripe fossil prices by Martill [194]). The permanence of the fossil material in regional institutions, on the other hand, such as museums and universities, has the potential to generate a more sustainable and lasting economy, with a greater and more equal distribution of income. Museums and geo- or palaeontological parks attract tourists, which help to support a network of establishments and people in the service sector, such as restaurants, hotels, gas stations, tourist guides, handicraft makers and retail outlets. In addition, they provide support for educational institutions, helping to train personnel and generate scientific and technological products. The simple withdrawal and international trade of fossils do not effectively contribute to the economic development of the region. Purchasing important fossils, either by private collectors or foreign researchers/museums, is depriving local museums of attractions that could potentially boost visitors and contribute to the local economy. The support of this type of activity keeps 'the colonized country' as an eternal exporter of commodities dependent on its 'colonizer' to provide specialized services and products.

Nevertheless, we do not criticize the commerce of fossil-bearing limestone nor the mining operations *per se*, as long as they comply with legal requirements and their environmental impact is addressed. Limestone mining is an important source of employment in many areas. Furthermore, many important fossils would not have been uncovered if it wasn't for commercial mining. Local institutions in both Brazil and Mexico regularly visit them in order to prevent significant fossils from being destroyed or accidentally sold as construction material. In Araripe, outreach activities are carried out involving quarry workers in order to raise awareness and encourage them to report these fossils [81].

## 7.3. Lack of interaction with local scientists can generate poor-quality research

Although the inclusion of local researchers is not required by law in Mexico, it is compulsory in Brazil (see §2.1). Local researchers have more specific knowledge about the geological context of the region, the co-occurrence of fossils, relevant studies that have been published in local or regional magazines (often in the local language), and other useful information, such as security/safety or socio-political issues. International collaboration can contribute significantly to the training of local researchers, which is one way to give back to the country of origin of the fossil material. In addition, local researchers who regularly interact with, or have even grown up and lived in, communities close to fossil sites are best placed to understand the economic and cultural needs of these communities, as well as to conduct outreach activities focused on the significance of the local fossils. Outreach activities are necessary to raise awareness and protect the heritage (see §6.4), and they can also result in new fossil discoveries by local people.

An example of a problem produced by non-cooperation with local researchers is the confusion that perpetuates around two spinosaurid dinosaurs from the Araripe Basin, Brazil, *Irritator* (figure 2*d*) and the strikingly similar form *Angaturama*. Both species were published in the same month, only a few days apart. *Irritator* was described exclusively by non-Brazilian researchers, based on the posterior portion of a skull, which was acquired from fossil traders [30]; and *Angaturama* was described by Brazilian researchers, based on the anterior portion of a skull, acquired through a donation from a private collection to the Universidade de São Paulo [195]. The situation caused some authors to speculate that the fossils belonged to the same individual [196,197], which was, much later, discarded [198]. Communication with local researchers and compliance with national legislation could have avoided this problem and resulted in a much more comprehensive work from the beginning. Moreover, this is another example of how the illegal collection of fossils can harm science. If the appropriate field data had been recorded during a legal and controlled collection, there would be no uncertainty about whether both fossil specimens belonged to the same individual animal.

## 7.4. Poor conduct in international collaborations erodes local expert trust

The *modus operandi* of scientific colonialism, in palaeontology and beyond, can generate widespread distrust and suspicion of foreign researchers who wish to collaborate with local researchers. Colonial scientific practices in palaeontology generate this distrust toward foreign parties, regardless of who is leading the project or which institution(s) the researcher(s) represent as they are equally seen as perpetuating extractive research at the expense of the local community. Conversely, local researchers who wish to partner with foreign colleagues are often looked upon with suspicion by other local researchers due to the bad image generated by third-party colonial practices. Overall, this situation prevents the progress of international scientific cooperation and hampers the development of local science and local researchers.

## 7.5. Fossil specimens that are difficult to access have a negative impact on local science development

When fossils are taken away to be housed in foreign institutions, they may become unavailable to local researchers and students, restricting the advancement of palaeontological research in their countries of origin. Visa-related issues [199] and reduced funding [14] are some of the major obstacles to international travel that are more likely to be faced by researchers in the Global South. Abysmal currency exchange rate differences routinely prevent Latin American researchers from travelling to access fossils in foreign collections. This is an especially serious problem with regard to access

to holotypes. While access to digital specimen data is becoming increasingly more available, certain types of research still require many fossils to be examined in person. Moreover, the difficulty of accessing the fossil material also implies that the research cannot be reproduced by peers who might be interested in replicating the study and/or conducting further tests. As such, removing fossils from their country of origin deepens the inequality between science produced in 'colonized' and 'colonizing' countries.

## 7.6. Poor-quality research can produce a large amount of dubious data

Sometimes, fossils can be collected without regard to associated taphonomic or stratigraphic information, especially when the collection is coordinated by commercial collectors. As a result, crucial information for the accurate understanding of that material is permanently lost. Some publications on Brazilian specimens examined in this study only vaguely state when the fossil was collected, instead of providing comprehensive geographical and geological context (see §4.2). We cannot discount that this might be intentional, as admitting that a Brazilian fossil was found after 1942 implies that a collection permit from ANM would have been legally required to collect it. The absence of permit information from a publication does not necessarily indicate that a particular team did not obtain the required permits. In fact, several local palaeontologists have also not reported the required collecting permits in their publications. In situations where fossil trading is permitted or fossil trafficking develops, fossils can even be artificially modified by fossil collectors or sellers to value them, as was the case with *Irritator challengeri* [30]. This results in poor-quality or non-reproducible research, and correcting these errors can require years of research, greatly delaying the advancement of science. The burden of this labour usually falls on researchers in the fossils' country of origin, as they are best placed to 'set the record straight' given their knowledge of the site, local literature and comparative specimens. This further exacerbates inequalities in global palaeontological research as it takes local researchers' time and energy away from other projects.

## 7.7. 'Fossil laundering'

In some cases, irregularly acquired fossils have been made to appear more ethically palatable to journals and the wider research community by adding a local researcher as a secondary author. In many instances of this, the local author is not affiliated to any research institution (e.g. [32,68,99,200,201] (figures 4 and 5, electronic supplementary material, tables S1 and S2). While we advocate for giving credit where it is due, this 'token authorship' represents the power dynamics at play and is a clear attempt at 'fossil laundering'. Amateur palaeontologists produce very valuable contributions to science often while working with their personal resources [202,203]. For example, in Coahuila, Mexico, the Paleontólogos Aficionados de Sabinas A.C. (Civil Association of Amateur Palaeontologists of Sabinas, figure 8c) established the Múzquiz Palaeontology Museum in 2005 [62]. Working with local dedicated amateurs and students is immensely important for reasons outlined in §6.4, and we do not criticize their deserved inclusion as co-authors in studies in which they took part, particularly as they are the group most likely to be excluded from publications more generally. Our criticism is instead directed towards the absence of true partnerships between foreign researchers and local scientific institutions. In many instances, this practice frequently creates ethical and, in some cases, legal problems and promotes the exploitation of local communities.

# 8. Historical examples of foreign collaborations in Brazil and Mexico

We are not advocating for a nationalistic approach to palaeontological research. International collaboration is an inherent part of the scientific endeavour and aids the development of local science. International collaborations are not a new phenomenon in Mexico or Brazil. In 1950, the Department of Palaeontology of Petróleos Mexicanos (PEMEX) was established, where Manuel Maldonado-Koerdell (of Mexican nationality) and Federico Bonet (of Spanish nationality) began the training of Mexican palaeontologists. Maldonado-Koerdell is considered the founder of the modern Mexican palaeontology community [204]. He promoted international cooperation as noted in his work 'Correlation of the Triassic formations of Central America, México and the United States' [205]. He had a great interest in Pan-Americanism and collaborated with Central American government agencies on projects related to palaeontology and mining (in Nicaragua) [204]. The palaeontologists

trained by Maldonado-Koerdell, were, in turn, pioneers in the study of other organisms. Agustín Ayala-Castañares was a pioneer of micropalaeontology in Mexico and had successful collaborations with several foreign researchers from Switzerland, the USA and Cuba [206–209]. His collaborations are also highlighted in the work 'Opportunities and challenges for Mexico-US cooperation in ocean sciences' [210].

Llewellyn I. Price, a Brazilian-born palaeontologist who worked with Alfred. S. Romer in the USA, teamed with a number of foreign researchers after returning to Brazil [211–214] and eventually became known as the 'founder of vertebrate palaeontology in Brazil'. Brazilian palaeontologist Carlos de Paula Couto, formerly Price's student, partnered with George Gaylord Simpson [215,216] and became the main specialist on fossil mammals in Brazil, further training generations of new local researchers. Both Llewellyn I. Price and Carlos de Paula Couto were prolific authors [217] and contributed greatly to the development of Brazilian science.

# 9. Small steps in the right direction and the repatriation challenge

Some research groups involved in scientific colonialism practices seem to be reviewing their approaches. Extended research on the Brazilian petrified wood stored at the Chemnitz Museum, Germany has been developed by Brazilian scientists in collaboration with German researchers [218–222] with funding provided by Brazil's Conselho Nacional de Desenvolvimento Científico e Tecnológico, CNPq (National Council for Development of Research and Technology). Despite this valuable collaboration, the irregularly acquired and exported specimens of fossil wood, including several holotypes, remain in Germany. Repatriation and restitution of fossils, especially ones that were acquired illegally, constitute key processes toward the decolonization of palaeontology by addressing the injustices under colonial or neocolonial contexts, as well as shifting the power of protecting heritage resources to the country of origin [223–226].

Several museums in the Global North are facing demands to repatriate fossils, and indeed other cultural and anthropological objects, and some countries making those demands have been successful [227,228]. One high-profile case that set a precedent for the application of national fossil laws to another country was that of the repatriation of two *Tarbosaurus bataar* specimens along with 16 other fossil specimens from the United States to Mongolia following a successful legal case in 2013 [229]. This success is due to the efforts of Bolortsetseg Minjin, a Mongolian palaeontologist who alerted the Mongolian authorities after seeing an advertisement for the auction of a *Tarbosaurus* specimen in New York. Neil Kelley, a palaeontologist based in the USA, started a petition on change.org to stop the auction [230], which attracted the attention of an American lawyer, Robert Painter, and Phillip Currie, a Canadian palaeontologist, who along with Minjin and Kelley provided expertise to show that these fossil specimens originated in Mongolia [231]. This paved the way for the repatriation of more than 30 Mongolian fossil specimens from the United States [232], as well as from France and South Korea [233].

More recently in 2019, in a case that draws parallels with the *Tarbosaurus* one, a French court ruled in Brazil's favour for the repatriation of 45 fossil specimens originating from the Araripe region [234]. After being alerted on Facebook to the online auction of one of the biggest and almost complete *Anhanguera santanae* specimens, Brazilian palaeontologists alerted the Brazilian Public Prosecutor's Office which immediately launched an investigation with the help of French authorities [87,234]. Another case of successful repatriation is that of Chinese fossils from Australia in 2008 thanks to the efforts of Australian palaeontologist John Long with the help of his Chinese collaborators who worked together with the Australian Federal Police following a request from China in 2004 [235]. A recently described, new species of spider, *Cretapalpus vittari* [236], was repatriated to Brazil only months after publication, together with 35 other undescribed arachnid specimens (figure 10) that were irregularly housed at the University of Kansas, USA [237]. Repatriation was done voluntarily at request of one of the authors of the study (M. R. Downen) after becoming more familiar with the issues involved with these specimens through information available on social media. These cases clearly highlight the importance of local expertise, collaborative networks and social media in uncovering and fighting the illicit trafficking of fossils.

As previously mentioned, a large number of Brazilian fossils have illegally ended up in foreign collections in Germany and Japan. At least 12 holotypes of Araripe fossil vertebrates and invertebrates are illegally housed in museums and private collections in Japan (e.g. Collection Masayuki Murata) (see electronic supplementary material, table S2). At least 90 Araripe holotypes were smuggled and ended in German museums (see §4.2 and electronic supplementary material, tables S2 and S3), mostly

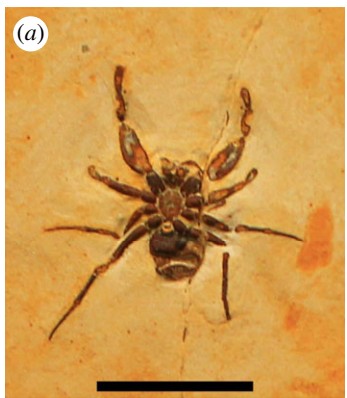 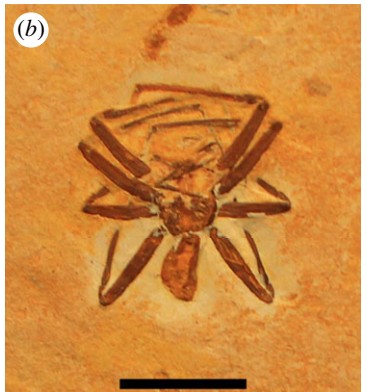 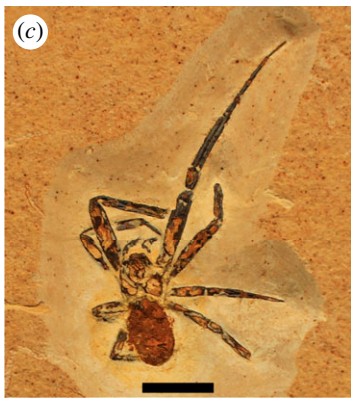

**Figure 10.** Representative specimens of 36 recently repatriated spiders from Araripe Basin, Brazil. (*a*) holotype of *Cretapalpus vittari*. (*b*, *c*) unidentified spiders, temporarily uncatalogued. Previously stored at the University of Kansas Natural History Museum, they are now reposited at Museu de Paleontologia Plácido Cidade Núvens at Universidade Regional do Cariri (URCA), Santana do Cariri, Ceará. Scale bar represents 5 mm.

in Stuttgart, Karlsruhe and Berlin. Brazil faces great difficulty in repatriating these fossils. Historically, these countries have been the least in favour of the 1970 UNESCO Convention [238] and only ratified the Convention themselves in the 2000s. Guidelines for the restitution of objects under 'colonial contexts' in Germany were only developed in 2019 but are not legally binding and do not provide any reliable legal framework within which legal claims can be made [239]. In addition, any object that was imported into Germany before 2007 does not fall under the protection of the law drafted in Germany after the ratification of the UNESCO 1970 Convention [240]. That said, Chile has successfully retrieved fossils stored in German collections, with the assistance of a German museum [227]. Conversely, the repatriation of the *Psittacosaurus mongoliensis* specimen, smuggled out of China and purchased by the same German museum after changing 'owners' several times, has been largely unsuccessful [241,242] and as of 2020, still had not been repatriated [243]. German museums recently agreed to the repatriation of several archaeological artefacts acquired during the colonial times [239], including Benin bronzes to Nigeria [244]. We expect that this will in the future be extended to palaeontological specimens as well.

# 10. Ways forward

## 10.1. Advances in technology and information sharing

Technology can be an ally in the process of decolonizing palaeontology. Museums and scientific institutions can share replicas, CT-scans, photogrammetry data and three-dimensional prints of important fossils, allowing information to be exchanged more widely and more equitably [245]. An international party working on Permian fossils in northeastern Brazil is already using this approach. Newly collected vertebrate fossils are reposited at Universidade Federal do Piauí, in Teresina, Brazil, while casts, photogrammetric models and CT-scans of the fossils are shared among various partner institutions overseas [246–249]. Researchers should consider making their CT-scan data available in an online repository such as MorphoBank or MorphoSource, which would facilitate equitable data sharing and reproducibility. Ideally, fossils, especially holotypes, should be repatriated to their country of origin. Fossil repatriation is generally perceived negatively by the museums that face these demands, who foresee large costs associated with the shipment of the material, as well as the loss of prominent specimens that may attract the public or provide valuable research data. However, the process of repatriation has enormous potential to establish new partnerships in the countries of origin, especially when new technologies, such as those mentioned above, are embraced.

## 10.2. Recommendations to journal editors and reviewers

Palaeontology-oriented journals, and those that routinely publish palaeontological studies, should adopt more rigorous guidelines regarding research ethics and the legal status of the fossils featured

in their articles. Ideally, journals should require authors to report both collecting and exportation permits where applicable, i.e. when a study is based on fieldwork and when fossils are not reposited in their countries of origin. Some journals with a history of publishing questionably acquired fossils (see electronic supplementary material, tables S2 and S3) seem to be moving in the right direction. *Palaeontology* and *Papers in Palaeontology* have both recently updated their guidance to authors, which now states that all manuscripts should include 'clear provenance information' and that 'samples should always be collected and exported in accordance with relevant permits and local laws, and in a responsible manner' [250]. Both journals also now require specimens to be 'deposited in a recognized museum or collection to permit free access by other researchers in perpetuity' [250]. Though this guidance does not include any specific examples of where permits would be required, or where authors might go to find more information, it is certainly a step in the right direction. *Current Biology* and other Cell Press journals, which frequently publish palaeontological studies, advocate for transparency and accessible reporting [251], and as of January 2021, allow authors to attach an inclusion and diversity statement in an attempt, among others, to curb the amount of scientific colonialism present in academia [252]. A recent study describing a new Chinese pterosaur fossil published in *Current Biology* used this opportunity to highlight that the author list of this publication included 'contributors from the location where the research was conducted who participated in the data collection, design, analysis and/or interpretation of the work' [253].

Other palaeontological journals apparently have strict policies regarding ethics and legality, but unfortunately do not put them into practice. *Cretaceous Research*, for example, states in its guide for authors that 'Fossil material of uncertain or dubious provenance will not be accepted for publication in *Cretaceous Research*. This includes material currently housed in museum collections which lack detailed field collecting records, and/or which provenance cannot be definitively ascertained with certainty' [254]. This journal, however, notoriously continues to publish numerous fossils of highly questionable origin from regions such as Araripe [255–258] including '*U. jubatus*' [26], finally removed after complaints from other researchers and social media pressure. *Cretaceous Research* also continues to publish articles on amber from Myanmar (54 publications as of October 2021), despite many other palaeontology journals placing restrictions on the publication of this material for ethical and legal reasons [130–132] (see discussion in §5).

In both Brazil and Mexico, fossil trade or export without a permit is illegal. Manuscripts involving fossils from these countries that only include vague statements involving the acquisition of a specimen or that cannot provide full information regarding how the fossil was obtained, should not be considered for publication. Brazilian fossils collected after 1942 and stored in foreign collections should be regarded with high suspicion. Authors who claim that a Brazilian fossil in a foreign collection was obtained before 1942 should be able to produce evidence of this. We are aware that Brazil's ANM authorized the legal exportation of some Brazilian fossils after this date (e.g. *Prionosuchus plummeri* specimens stored at the Natural History Museum in London, UK [259]), but these are very rare cases that involve non-holotype specimens. Any Brazilian holotype stored in a foreign collection that was collected after 1990 represents a violation of Brazilian law, as demonstrated above. Authors must obtain and provide proper documentation from ANM and MCTI that demonstrates that these fossils were collected and exported legally. Fossils that are kept in private collections and not in research institutions, should also not be considered for publication, especially when involving new taxa. Mexican fossils having an INAH registration number are not necessarily reposited in museums or universities because this government agency also registers private collections. Moreover, in order to encourage registration, INAH does not ask questions regarding provenance or authenticity of the fossils. Authors that wish to publish Mexican fossils registered by INAH should provide documentation proving that the fossil in question is available to scientists in a research institution and not part of a private collection. As mentioned above, some foreign research groups who usually work on these fossils openly advocate breaking local laws and regulations [10,11,177]. We strongly recommend that editors refuse to publish studies on these fossils unless their legal status has been clearly demonstrated with supporting documentation by the authors. In cases where journal editorial policies are lacking or inadequate, we recommend that reviewers go beyond the requirements of the journal to ask these questions and demand documentation proactively. While it is imperative that research complies with the national legislature, there is no legal requirement to engage in work that is ethical or abstain from parachute science. We, therefore, must also recommend that editors, reviewers and authors be mindful of the implications of research that is not equitable.

## 10.3. Recommendations to local governments and authorities

Governments should strictly enforce current laws and regulations, and regularly review laws in direct consultation with experts and stakeholders. In August 2020, a bill was passed in the Nuevo León Congress that redefined the material from Vallecillo (in the Sabinas Basin), collected for over two decades, as 'unusual engravings in carbonate limestone'. One of the main concerns, as reported by journalists at the time [260] and voiced by the Council of Palaeontology of INAH [261], is that the new law now facilitates the open trade of fossils extracted from the Vallecillo quarry, bypassing fossil protection legislations [260,261]. In order to make our study more accessible to local authorities and policy makers, we have translated the text into Spanish and Portuguese (see supplementary electronic material, Translation S1 and S2).

However, due to the asymmetry of power between the governments of formerly colonized countries and major colonial powers of the Global North, the duty of preventing fossil smuggling poached from countries with robust heritage laws must also be taken up by the countries where these fossils are ultimately purchased. Efforts to counter smuggling of cultural and national heritage objects, including fossils and protected wildlife, have been undertaken in some countries of the Global North, notably Canada and the USA [229,262], but this effort has largely focused on singular high-profile items (e.g. dinosaur skeletons) rather than general trade in contraband heritage objects. Other countries, such as Germany, have previously been referred to as being one of the centre of international trade of illicit antiquities [263]. More robust adherence to international conventions on smuggling of heritage objects by destination countries is a critical piece of any effort to bolster local fossil protection legislation.

## 10.4. Recommendations to research institutions, funding agencies and reviewers

Palaeontology is a science that captures wide public interest [264,265]. Colonial science practices repeatedly carried out by some palaeontologists can evolve into a negative public perception toward any academic institution or museum that tolerates them. Complying with local regulations is not only a logical step before undertaking research on internationally sourced specimens, but should also be an expected ethical practice for palaeontologists and scientists in general. Some professional palaeontological societies now have a code of conduct that their members must comply with, which addresses this specific point [266,267]. Museums and universities should advise and support their staff on expected ethical and legal conduct in low-income or resource-poor countries in order to avoid conducting colonial science practices. Universities should also consider including science history and ethics courses as part of their undergraduate and graduate programmes to ensure that the future generation of palaeontologists receive adequate training on these issues within palaeontology (as well as the geo- and biological sciences more generally). Funding agencies should make it compulsory for applicants to demonstrate that they will comply with the laws of the countries they wish to carry out research in. Failure to do so should result in the funding being terminated. Funders should also have a requirement that equitative, institutionalized cooperation with local counterparts be facilitated in a way that benefits both parties involved in the collaboration. As with manuscript reviews, grant reviewers should consider pre-emptively scrutinizing proposed partnerships and international fieldwork irrespective of whether the funder currently requires this oversight. Reviewers should also be mindful of the ethical implications of proposed partnerships and outcomes, outside of legal requirements.

Museums share a burden of responsibility for colonial practices that is difficult to ignore [13,29]. It is hard to imagine that nearly a hundred Araripe holotype fossils (and presumably an even larger number of non-type fossils) made it illegally to foreign museums without the knowledge or even support from their respective curators. As seen above, there are several cases in which authors openly admit that fossils were purchased. Details of purchases are even recorded on museum labels within collections (figure 11a), and there are even cases in which Araripe fossils are being sold by the museum gift shops (figure 11b). Museums should adopt strict policies regarding the reception of specimens in their collections. Banning the admission of fossils with dubious provenance data or from countries that prohibit their exportation, as is the case of Brazil and Mexico, should be embraced as a formal policy.

## 11. Conclusion

Our study provides an extensive overview of colonial science practices that deepen global inequalities in palaeontological research. Though our investigation focuses on Brazil and Mexico, these practices occur

(a)

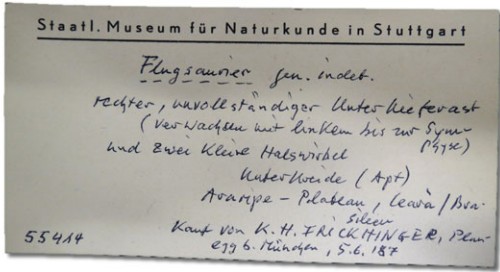

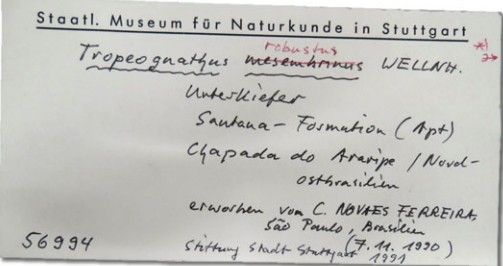

(b)

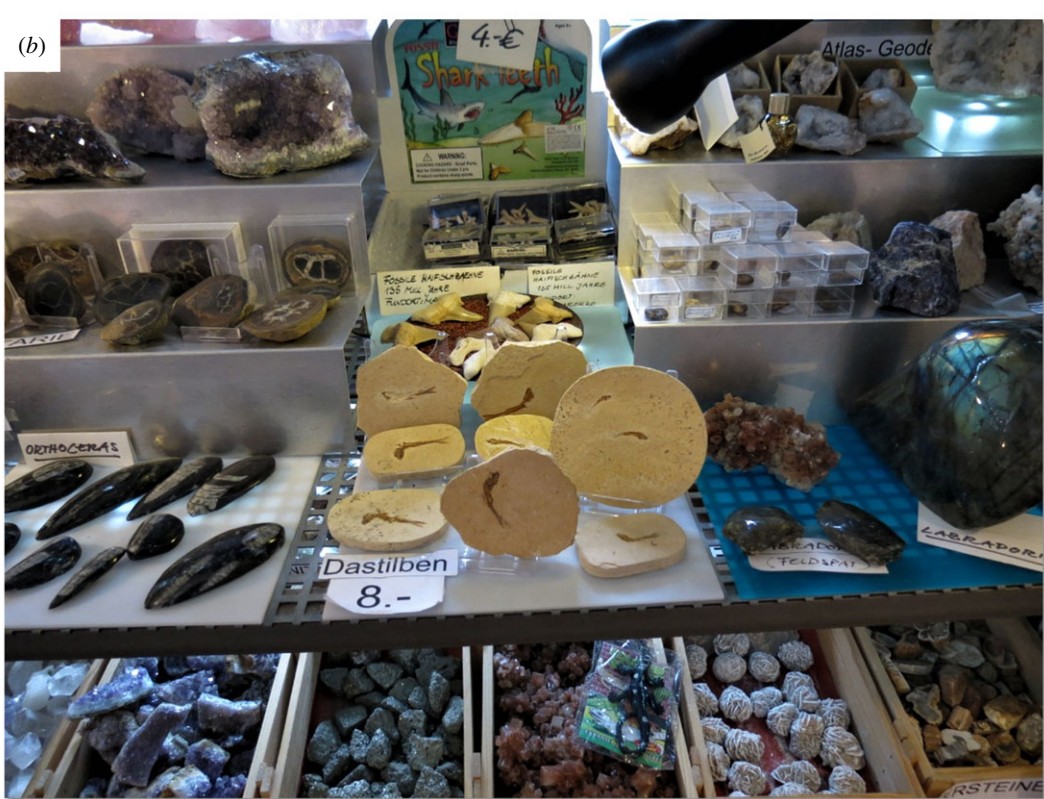

**Figure 11.** (a) acquisition of Araripe fossils by Staatliche Museum für Naturkunde Stuttgart. SMNS 58022 holotype of the dinosaur *Irritator challengeri*, (label says 'purchased from M. Kandler 1991'); SMNS 55 414 pterosaur (indeterminate genus), (label says 'purchased from K. H. Frickhinger, Planegg in Munich, 5.6.187' [*sic*]); 82 001 pterosaur (indeterminate genus), (label says 'purchased from K. H. Frickhinger Planegg in Munich, 5.6.1987, together with 55 404–55 415 for the price'); 56 994 pterosaur *Tropeognathus robustus*, (label reads 'acquired from C. Novaes Ferreira, São Paulo, Brazil (7.11.1990)'. (b) Araripe fishes (*Dastilbe* sp.) being sold at a souvenir shop in Staatliche Museum für Naturkunde Karlsruhe in 2011. Commerce and exportation of fossils has been forbidden in Brazil since 1942 (see §2 and box 3).

in many other countries, as shown in the numerous cases presented above of fossil smuggling, forgery and even blatant disregard for national laws. Museums, universities and funding agencies must avoid facilitating research or researchers that are involved in colonial scientific practices, especially when there are signs of violation of local laws and regulations, such as the illegal purchase and export of

fossils. It is equally important that scientific journals mandate that authors provide the required research and exportation permits alongside their manuscripts, and refuse to publish research that is produced through unethical and irregular activities, such as the cases detailed above. Foreign researchers must respect local laws and regulations, and engage in constructive, ethical and equitable partnerships. Poor research conduct that cuts both ethical and legal corners results in the erosion of trust in experts, dubious and non-reproducible research and increased difficulty in accessing important fossil specimens. The extractive history of colonialist palaeontology cannot be rewritten, but we can forge a new path based on respectful cooperation that mutually benefits both local and foreign institutions, as well as the local communities that remain the custodians of their palaeontological heritage.

Data accessibility. Supplementary tables, appendices and translations of the article (Spanish and Portuguese) are available at Zenodo https://doi.org/10.5281/zenodo.6358846 [268].

Authors' contributions. J.C.C.: conceptualization, investigation, methodology, project administration, visualization, writing—original draft, writing—review and editing; N.B.R.: data curation, formal analysis, investigation, methodology, visualization, writing—original draft, writing—review and editing; A.M.G.: investigation, methodology, writing—original draft, writing—review and editing; E.M.D.: investigation, writing—original draft, writing—review and editing; F.L.P.: investigation, visualization, writing—original draft, writing—review and editing; O.R.R.F.: investigation, writing—original draft, writing—review and editing; M.A.F.S.: investigation, writing—original draft, writing—review and editing; R.A.R.-R.: visualization, writing—original draft; A.Y.M.-M.: visualization, writing—original draft, writing—review and editing; S.G.-M.: writing—original draft, writing— review and editing; R.A.M.B.: investigation, writing—review and editing; F.J.L.: investigation, writing—original draft; J.D.P.: writing—original draft, writing—review and editing.

All authors gave final approval for publication and agreed to be held accountable for the work performed therein.

Competing interests. We declare we have no competing interests.

Funding. N.B.R. was supported by Deutsche Forschungsgemeinschaft (KI 806/17-1).

Acknowledgements. We would like to express our gratitude to law officers that investigate and protect fossil heritage from illicit trafficking, to the journalists and communicators that expose these issues and raise awareness, and to the general public that talk about this topic and keep it alive on social media and help to add pressure to the individuals and institutions responsible for purchasing and withholding this heritage. This manuscript had input from other colleagues who preferred to remain anonymous. We appreciate the help provided by Alberto Blanco Piñón through our discussions on the Sabinas fossils as well as the photographs he provided. We are thankful to Felisa Aguilar Arellano (Consejo de Paleontología – INAH) for her valuable help with Mexican fossil laws. Chico Camargo supported us in extracting the Twitter data in this study. Our thanks also go to: Paolo Schirolli (Museo Civico di Scienze Naturali di Brescia) and Lorenzo Marchetti (Museum für Naturkunde Berlin) who provided information regarding Italian laws, Jeff Liston (Royal Tyrrell Museum of Palaeontology) and John Long (Flinders University) for providing information on previous repatriations of fossils, and Edenilce P. Batista (Universidade Regional do Cariri) for information on Araripe plants. The photograph of *Tetrapodophis amplectus* (figure 2*f*) was provided by Michael Caldwell (University of Alberta), images in figure 1*e,f* are courtesy of El Norte—Grupo REFORMA, photograph of La Mula Quarry (figure 3*b*) was provided by Selene Velázquez, and the art in figure 8*d* was created by Saulo Daniel Ferreira Pontes. The manuscript also greatly benefited from comments by Sarah Greene (University of Birmingham), Shazia Kurmoo (Ministry of Foreign Affairs, Mauritius), reviewer Jean-Noel Martínez (Universidad Nacional de Piura) and an anonymous referee. We are also grateful to Alexandra Elbakyan and the Sci-Hub project for providing access to several publications that were necessary to this study. Finally, our very special *obrigado* to Matthew R. Downen and the University of Kansas Natural History Museum for collaborating with the repatriation of 36 fossil spiders to Brazil.

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
