## [Peer Review File · Royal Society Open Science]

Review History

RSOS-210898.R0 (Original submission)

Review form: Reviewer 1 (Michelle A. North)

Is the manuscript scientifically sound in its present form?

Yes

Are the interpretations and conclusions justified by the results?

Yes

Is the language acceptable?

Yes

Do you have any ethical concerns with this paper?

No

Have you any concerns about statistical analyses in this paper?

No

Recommendation?

Accept with minor revision (please list in comments)

Comments to the Author(s)

Please see the attached review report with my comments (Appendix A).

Review form: Reviewer 2 (Jean-Noël Martinez)

Is the manuscript scientifically sound in its present form?

Yes

Are the interpretations and conclusions justified by the results?

Yes

Is the language acceptable?

Yes

Do you have any ethical concerns with this paper?

No

Have you any concerns about statistical analyses in this paper?

No

Recommendation?

Accept with minor revision (please list in comments)

Comments to the Author(s)

A lucid, well written, abundantly documented paper, Very necessary in this moment... Only minor modifications indicated directly in the text (attached pdf (Appendix B)).

Decision letter (RSOS-210898.R0)

Dear Dr Cisneros

The Editors assigned to your paper RSOS-210898 "Digging deeper into colonial palaeontological practices in modern day Brazil and Mexico" have now received comments from reviewers and would like you to revise the paper in accordance with the reviewer comments and any comments from the Editors. Please note this decision does not guarantee eventual acceptance.

Please submit your revised manuscript and required files (see below) no later than 21 days from today's (ie 14-Oct-2021) date. Note: the ScholarOne system will 'lock' if submission of the revision is attempted 21 or more days after the deadline. If you do not think you will be able to meet this deadline please contact the editorial office immediately.

Kind regards,
Royal Society Open Science Editorial Office
Royal Society Open Science
openscience@royalsociety.org
on behalf of Dr Miranda Lowe (Associate Editor) and Nick Pearce (Subject Editor)
openscience@royalsociety.org

Associate Editor Comments to Author (Dr Miranda Lowe):

Thank you for your patience while we sought reviewers for this paper - regrettably, this took a lot longer than usual (and the editors have had to approach a very large number of potential reviewers to receive the two reports we now have). In any case, two reviewers have offered a number of comments that you need to address before we can consider the paper ready for publication - reviewer 1 in particular has a number of concerns regarding, for example, the rigor of your description of your methodologies. This needs to be addressed in any revision. Please ensure that you provide not only a tracked-changes iteration of your paper with the revision but also a clear point-by-point response document that delineates the changes you have made. Thanks for your support and good luck with the revision.

Reviewer comments to Author:

Reviewer: 1

Comments to the Author(s)

Please see the attached review report with my comments (attached pdf: "RSOS-210898 reviewer report.pdf")

Reviewer: 2

Comments to the Author(s)

A lucid, well written, abundantly documented paper, Very necessary in this moment... Only minor modifications indicated directly in the text (attached pdf: "RSOS-210898_Proof REVIEW JNM.pdf").

===PREPARING YOUR MANUSCRIPT===

===PREPARING YOUR REVISION IN SCHOLARONE===

-- If you have uploaded ESM files, please ensure you follow the guidance at <https://royalsociety.org/journals/authors/author-guidelines/#supplementary-material> to include a suitable title and informative caption. An example of appropriate titling and captioning may be found at https://figshare.com/articles/Table_S2_from_Is_there_a_trade-off_between_peak_performance_and_performance_breadth_across_temperatures_for_aerobic_sc_ope_in_teleost_fishes_/3843624.

Author's Response to Decision Letter for (RSOS-210898.R0)

See Appendix C.

RSOS-210898.R1

Review form: Reviewer 1 (Michelle A. North)

Is the manuscript scientifically sound in its present form?

Yes

Are the interpretations and conclusions justified by the results?

Yes

Is the language acceptable?

Yes

Do you have any ethical concerns with this paper?

No

Have you any concerns about statistical analyses in this paper?

No

Recommendation?

Accept as is

Comments to the Author(s)

Review report for RSOS-210898.R1

Title: Digging deeper into colonial palaeontological practices in modern day Mexico and Brazil

The authors have made extensive changes to their manuscript based on the first round of reviewer comments, and this hard work has certainly paid off. The Supplementary files are now extremely accessible and useful, and I hope that future researchers wanting to collaborate with locals make use of the list relevant experts and institutions the authors have provided. I sincerely appreciate the translated column headings, and feel that the additional information provided adds a lot of value to these tables. It is a pity the revised figures were not included in the manuscript, but I am sure the font will be more legible (based on the authors' responses).

I look forward to reading the final, published version of this manuscript, and with the authors well on their future work on this issue. I hope this paper continues to improve awareness of this issue and paves the way to more repatriation of fossils as well as greater collaboration with local researchers.

Review form: Reviewer 2 (Jean-Noël Martinez)

Is the manuscript scientifically sound in its present form?

Yes

Are the interpretations and conclusions justified by the results?

Yes

Is the language acceptable?

Yes

Do you have any ethical concerns with this paper?

No

Have you any concerns about statistical analyses in this paper?

No

Recommendation?

Accept with minor revision (please list in comments)

Comments to the Author(s)

Excellent paper, well documented and very necessary in this moment. I found very few necessary and purely formal corrections that are indicated in the attached file (Appendix D).

Decision letter (RSOS-210898.R1)

Dear Dr Cisneros

On behalf of the Editors, we are pleased to inform you that your Manuscript RSOS-210898.R1 "Digging deeper into colonial palaeontological practices in modern day Mexico and Brazil" has been accepted for publication in Royal Society Open Science subject to minor revision in accordance with the referees' reports. Please find the referees' comments along with any feedback from the Editors below my signature.

Please submit your revised manuscript and required files (see below) no later than 7 days from today's (ie 24-Jan-2022) date. Note: the ScholarOne system will 'lock' if submission of the revision is attempted 7 or more days after the deadline. If you do not think you will be able to meet this deadline please contact the editorial office immediately.

on behalf of Dr Miranda Lowe (Associate Editor) and Nick Pearce (Subject Editor)
openscience@royalsociety.org

Subject Editor Comments to Author (Professor Nick Pearce):

Comments to the Author:

We shall be pleased to publish this important paper, subject to the minor final revisions recommended by the second reviewer. Many thanks for choosing to submit to the Science, Policy and Society section of RSOS.

Reviewer comments to Author:

Reviewer: 1

Comments to the Author(s)

Review report for RSOS-210898.R1

Title: Digging deeper into colonial palaeontological practices in modern day Mexico and Brazil

The authors have made extensive changes to their manuscript based on the first round of reviewer comments, and this hard work has certainly paid off. The Supplementary files are now extremely accessible and useful, and I hope that future researchers wanting to collaborate with locals make use of the list relevant experts and institutions the authors have provided. I sincerely appreciate the translated column headings, and feel that the additional information provided adds a lot of value to these tables. It is a pity the revised figures were not included in the manuscript, but I am sure the font will be more legible (based on the authors' responses).

I look forward to reading the final, published version of this manuscript, and with the authors well on their future work on this issue. I hope this paper continues to improve awareness of this issue and paves the way to more repatriation of fossils as well as greater collaboration with local researchers.

Reviewer: 2

Comments to the Author(s)

Excellent paper, well documented and very necessary in this moment. I found very few necessary and purely formal corrections that are indicated in the attached file.

===PREPARING YOUR MANUSCRIPT===

one version should clearly identify all the changes that have been made (for instance, in coloured highlight, in bold text, or tracked changes);

If you have been asked to revise the written English in your submission as a condition of publication, you must do so, and you are expected to provide evidence that you have received language editing support. The journal would prefer that you use a professional language editing

service and provide a certificate of editing, but a signed letter from a colleague who is a proficient user of English is acceptable. Note the journal has arranged a number of discounts for authors using professional language editing services (<https://royalsociety.org/journals/authors/benefits/language-editing/>).

===PREPARING YOUR REVISION IN SCHOLARONE===

-- If you are requesting an article processing charge waiver, you must select the relevant waiver option (if requesting a discretionary waiver, the form should have been uploaded, see 'File upload' above).

-- If you have uploaded any electronic supplementary (ESM) files, please ensure you follow the guidance at <https://royalsociety.org/journals/authors/author-guidelines/#supplementary-material> to include a suitable title and informative caption. An example of appropriate titling and captioning may be found at https://figshare.com/articles/Table_S2_from_Is_there_a_trade-off_between_peak_performance_and_performance_breadth_across_temperatures_for_aerobic_scope_in_teleost_fishes_/3843624.

Author's Response to Decision Letter for (RSOS-210898.R1)

See Appendix E.

Decision letter (RSOS-210898.R2)

Dear Professor Dunne,

I am pleased to inform you that your manuscript entitled "Digging deeper into colonial palaeontological practices in modern day Mexico and Brazil" is now accepted for publication in Royal Society Open Science.

on behalf of Dr Miranda Lowe (Associate Editor) and Nick Pearce (Subject Editor)
openscience@royalsociety.org

Appendix A

Review report for RSOS-210898

Title: Digging deeper into colonial palaeontological practices in modern day Brazil and Mexico

This is a fascinating paper about scientific colonialism in palaeontology, with a specific focus on two fossil rich sites in Brazil and Mexico. The authors clearly describe the problem, the legal frameworks, several cases, as well as providing clear examples of 'myths' that they rebut and options for improving the current situation. I think this manuscript is an extremely valuable contribution to the scientific literature and to 'general' knowledge on palaeontology.

General comments:

Firstly, I would like to say that I really enjoyed reading this manuscript. The 'story' is interesting, and the way it has been presented is gripping. It reads like a novel that one does not want to put down – well done. It is also a very important topic, and clearly the authors have extensive experience in this field and with this issue. This comes across in the way it has been presented. However, I do have some relatively minor comments, and a more substantial one.

It is highly irregular to only include a description of the methods in the supplementary information. If a literature review is used, it should be described (even if very briefly) in the main text of the document, so that readers have some idea of the basis of the data used and the conclusions drawn. And even when I access the supplementary material, all it contains are hyperlinks to a web based repository that then autodownloads a zip file of further tables... This is not how one reports methods, but is rather an admirable way of presenting the data used. Somewhere, at least in the Supplementary information but preferably in the main text, there has to be a written description of the methods used to obtain the data upon which part of this study has been based. This should also include a brief description of the Twitter analytics alluded to in the Acknowledgements.

For example, in Table S1, there is no description of how these papers were found (what search terms, what databases or platforms?), or of the content of this table (what does n/a mean for 'collecting permit'? Does it mean there was not one, it was not mentioned, or that the work described in that paper did not need a permit? Under 'Issues', what do the different numbers mean?). All of this information needs to be described somewhere. In addition to a description of the methods in the text, I would also recommend adding a 'read me' sheet to each of the supplemental data files, where you explain what the different columns and numbers etc mean.

On another note, regarding Table S1, I would like to see the titles of each of the publications, or a full reference, so that readers can go and find these different papers more easily.

Regarding the other Supplemental tables (I noticed this issue in Table S4 but it may be true for more of them), while I completely acknowledge that English is not the only language of science and is not the primary language of either of these regions or the authors of this paper, I would suggest having the table headings either in English or including their English translations in a 'read me' sheet. I realize that translating the data will be a pain, but if you are translating the main manuscript into Spanish and Portuguese anyway, then you might consider having different sheets in each (relevant) table for each of the three languages? Or simply provide an explanation in the first 'read me' sheet translating key terms (like "Coleção Científica, exposição, etc."). As an example, if the content of this specific column in Table S4 (Tipo) is confined to only two or three types of work, it might be easier for all different language readers to interpret if you rather set it up more like I have done in the screenshot below (I had to use Google translate, so I'm sure the headings are not quite correct). I appreciate that the issue of language is a tricky one, but for your paper to have maximum impact, it would be ideal for everything to be multilingual, or at least easy to interpret.

Nome	Instituição	Local	Estado	Scientific Collection exposure	Paleontological Collection	open air exposure	Tipo de coleção
Museu Nacional	UFRJ	Rio de Janeiro	RJ	x	x		
IGeo/Museu da Geodiversidade	UFRJ	Rio de Janeiro	RJ	x	x		
Museu de Ciências da Terra	CPRM	Rio de Janeiro	RJ	x	x		
Parque Paleontológico de São José do Itaboraí		Itaboraí	RJ		x		
Laboratório de Paleontologia (LAPA)	UERJ	São Gonçalo	RJ	x			
Laboratório de Paleontologia	UFES	Vitória	ES		x		
Museu de História Natural do Sul do ES		Jerônimo Monteiro	ES		x		
Museu de Ciências da Vida	UFES	Vitória	ES	x	x		
Museu Paleontológico de Pirópolis + CPPP	UFTM	Uberaba	MG	x	x		
Museu de Ciências Naturais	PUC	Belo Horizonte	MG	x	x		
Museu Peter Lund	Particular	Lagoa Santa	MG				
Museu de História Natural	UFMG	Belo Horizonte	MG	x	x		
Laboratório de Paleontologia e Macroevolução	IGC/UFMG	Belo Horizonte	MG	x			
Laboratório de Paleontologia	UFV	Viçosa	MG	x			
Laboratório de Paleontologia	UFU	Uberlândia	MG	x			
Museu de Paleontologia de Monte Alto	Municipal	Monte Alto	SP	x	x		
Museu de Paleontologia Pedro Candolo	Municipal	Uchoa	SP	x	x		

Specific comments:

For these comments, I refer to the page number from the document, and not the PDF page (for example, the abstract/summary is on page 3 which is the 4th page in the PDF. I will report this as Page 3. Also note that the PDF line numbers do not seem to fully correspond with the lines of text, so just estimate which line of text each line number I refer to is indicating.

Page 3 line16: "Common issues documented in these publications" are these issues truly documented by the publications you found (i.e., do the authors of those publications highlight the issues), or do you mean that they are issues that you found in these publications?

Page 3 line 23: "reposited" is not a commonly used word in English. I would suggest replacing with something like stored, housed, or kept. But I am not of this field, so if the term is commonly used in palaeontology then ignore my comment.

Page 4 line 19-23: I don't entirely understand this statement. I would recommend expanding what you mean by institutionalisation, and why it benefits colonial advancement

Page 5 line 30: consider rather using the term "while" here, rather than "whereas" (it isn't really a contrast to the previous phrase)

Page 5 line 40: Rather use the term "scientific reproducibility"

Page 6 line 26: Delete "being" (before "the USA")

Page 6 line 59: Rather change "previously" to "to notify the council beforehand"

Page 7 line 11-14: Consider making this sentence shorter (e.g., use "theoretically" instead of in theory, remove "in the first place", remove "properly"). This should make it easier to read, but if its still too long, then consider using punctuation or breaking it into two sentences.

Page 7 line 14: Delete "thus" before acknowledge

Page 7 line 19-26: This is a very long sentence. Rephrase to shorten, add punctuation, and consider rewriting as two sentences

Page 7 line 42: Is this not meant to be northEASTERN, rather than northwestern Mexico?

Page 8 line 14: this is the first time the literature review is mentioned. I would strongly suggest that the methods should be mentioned earlier so that the reader understands how you found these papers (refer to my General comments, above)

Page 8 line 18: It is not clear how Box 3 relates to this sentence

Page 8 line 19: Rather than "The majority of this output, I would rather state it something like "Most of this foreign-led research, ..." to make it clear that "this output" isn't referring to the 128 papers

Page 10 line 59: Here again, there needs to be some sort of description of the methods used to find these papers

Page 11 line 4: Does "them" (the 57.14%) refer to the foreign-led papers? Or of all the papers?

Page 12 line 24-26: I am not sure whether the evidence can prove that the foreign researchers avoided interaction. I would recommend moderating the language. One could say they neglected to include local scientists, or that this displays a disregard for local laws and expertise.

Page 12 line 37: Rather say that they "likely" represent only a small portion. Because you don't have the other numbers you cannot say for sure that they do represent only a small portion

Page 14 line 40: do you rather mean "lines of research" or "research practices"?

Page 14 line 42: "metropolis" isn't quite the correct word. I think you mean the colonising countries?

Page 14 line 49: It does not appear to solely be the absence of legislation, seeing as both the case study countries have adequate legislation; it appears to be more about the absence of enforced legislation

Page 20 line 14: How does the sale or purchase of fossils differ from the production of stone products (e.g., for paving or kitchen counters) from quarries that have fossil deposits? I am sure I have seen both the above with little fossil structures in them. And many finds occur in these quarries, which I assume are commercial entities? Without these commercial quarries I assume many fossil deposits would not be revealed. I am not a paleontologist at all, so excuse me if this is a very 'lay' question, but I wonder if this is something you could mention/discuss somewhere

Page 26 line 11: Would it not be possible to develop a framework somewhat akin to PRISMA regarding a checklist of items every paper about fossils should report (e.g., permit information, detailed description of the site, geology etc., current location of the fossils)? And it would be ideal if proposed research protocols were published for peer review before the research is actually undertaken, so that any gaps (e.g., the acquisition of permits) can be highlighted before the study is conducted.

Page 27 line 23-24: What do you propose as a solution for these illegal fossils? Should they therefore not be studied, or if they are to be studied and published, should the specimens be repatriated or at least moved to public collections? My concern is the consequence if highly valuable, unique

specimens fall in this category, then they will not make it to the scientific literature... should the publications simply be open about the fact that they are studying illegal specimens?

Page 51: In the caption of Figure 2, Frey et al 2017 is not referenced as a number the way all other references have been done.

Page 53 Figure 4: I recommend making this figure just a bit larger (fit the width of the page) so that the font is more legible. Also make the currently grey font colour black to improve contrast

Page 58 Figure 9: See whether it is possible to make this figure bigger, to fit the width of the page. I think the font size will be better then, and more legible

Appendix B**ROYAL SOCIETY
OPEN SCIENCE****Digging deeper into colonial palaeontological practices in
modern day Brazil and Mexico**

Journal:	Royal Society Open Science
Manuscript ID	RSOS-210898
Article Type:	Science, Society and Policy
Date Submitted by the Author:	03-Jun-2021
Complete List of Authors:	Cisneros, Juan; Universidade Federal do Piauí, Museu de Arqueologia e Paleontologia Raja, Nussaibah; Friedrich-Alexander-Universität Erlangen-Nürnberg, Ghilardi, Aline; Universidade Federal do Rio Grande do Norte, Departamento de Geologia Dunne, Emma; University of Birmingham, School of Geography, Earth and Environmental Sciences Pinheiro, Felipe; Universidade Federal do Pampa - Campus Sao Gabriel, Regalado Fernandez, Omar Rafael; University College London, Earth Science Department Sales, Marcos; Instituto Federal de Educação Ciência e Tecnologia do Ceará Rodríguez de la Rosa, Rubén; Universidad Autónoma de Zacatecas, Unidad Académica de Ciencias Biológicas-Unidad Académica de Ciencias de la Tierra Miranda-Martínez, Adriana; Universidad Nacional Autónoma de México, Departamento de Biología Evolutiva, Facultad de Ciencias González-Mora, Sergio; Universidad Nacional Autónoma de México, Museo de Paleontología, Departamento de Biología Evolutiva, Facultad de Ciencias Bantim, Renam; Universidade Regional do Cariri, Laboratório de Paleontologia, Departamento de Ciências Biológicas Lima, Flaviana; Universidade Federal de Pernambuco, Laboratório de Paleobiologia e Microestruturas, Centro Acadêmico de Vitória Pardo, Jason; University of Calgary, Department of Comparative Biology and Experimental Medicine
Subject:	palaeontology < BIOLOGY, Palaeontology < EARTH SCIENCES, taxonomy and systematics < BIOLOGY
Keywords:	scientific colonialism, parachute science, research ethics, illegal fossil trade, palaeontological heritage, Latin America
Subject Category:	Science, Society and Policy

Digging deeper into colonial palaeontological practices in modern day Brazil and Mexico

Juan Carlos Cisneros¹, Nussaïbah B. Raja², Aline M. Ghilardi³, Emma M. Dunne⁴, Felipe L. Pinheiro⁵, Omar Rafael Regalado Fernández⁶, Marcos A. F. Sales⁷, Rubén A. Rodríguez-de la Rosa⁸, Adriana Y. Miranda-Martínez⁹, Sergio González-Mora¹⁰, Renan A. M. Bantim¹¹, Flaviana J. de Lima¹², Jason D. Pardo¹³

RRH: PALEONTOLOGICAL COLONIALISM IN BRAZIL AND MEXICO

¹ Museu de Arqueologia e Paleontologia, Universidade Federal do Piauí (UFPI), Teresina, PI, 64049-550, Brazil, <https://orcid.org/0000-0001-6159-1981>

² GeoZentrum Nordbayern, Department of Geography and Geosciences, Friedrich-Alexander University Erlangen-Nürnberg, Loewenichstr. 28, 91054 Erlangen, Germany, <https://orcid.org/0000-0002-0000-3944>

³ Departamento de Geologia, Universidade Federal do Rio Grande do Norte (UFRN), Natal, Brazil, <https://orcid.org/0000-0001-9136-0236>

⁴ School of Geography, Earth and Environmental Sciences, University of Birmingham, Edgbaston, Birmingham, B15 2TT, UK, <https://orcid.org/0000-0002-4989-5904>

⁵ Laboratório de Paleobiologia, Universidade Federal do Pampa, São Gabriel, Brazil

⁶ Earth Science Department, University College London, London, United Kingdom, <https://orcid.org/0000-0002-6247-6181>

⁷ Instituto Federal de Educação, Ciência e Tecnologia do Ceará (IFCE) - Campus Acopiara, Ceará, Brazil

⁸ Unidad Académica de Ciencias Biológicas-Unidad Académica de Ciencias de la Tierra, Universidad Autónoma de Zacatecas, Calzada Solidaridad, S/N, Campus II, C.P. 98060, Zacatecas, Mexico. <https://orcid.org/0000-0002-7219-1550>

⁹ Departamento de Biología Evolutiva, Facultad de Ciencias, Universidad Nacional Autónoma de México, Ciudad Universitaria, 04510 Ciudad de México, México.

¹⁰ Museo de Paleontología, Departamento de Biología Evolutiva, Facultad de Ciencias, Universidad Nacional Autónoma de México, Ciudad Universitaria, 04510 Ciudad de México, México, <https://orcid.org/0000-0001-9709-2033>

¹¹ Laboratório de Paleontologia, Departamento de Ciências Biológicas, Universidade Regional do Cariri. Rua Coronel Antônio Luís, 1161, Pimenta, Crato, Ceará, Brazil, <https://orcid.org/0000-0003-4576-0989>

¹² Laboratório de Paleobiologia e Microestruturas, Centro Acadêmico de Vitória - Universidade Federal de Pernambuco (CAV/UFPE), R. Alto do Reservatório - Alto José Leal, Vitória de Santo Antão, Pernambuco, Brazil, <https://orcid.org/0000-0001-8602-6508>

¹³University of Calgary, Calgary, Alberta, Canada T2N 4N1

**Author contributions**

JCC, NBR, AMG and EMD conceived and designed the project. JCC, AMG, FLP, ORRF, MAFS,
AYM, SGM, RAMB, RARR and FJL compiled the data in the tables. NBR, JCC and FLP made the
figures. All authors contributed to and approved the manuscript.

Summary

Scientific practices stemming from colonialism, whereby middle- and low-income countries supply data for high income countries and the contributions of local expertise are devalued, is still prevalent today in the field of palaeontology. In response to these unjust practices, countries such as Brazil and Mexico adopted protective laws and regulations during the 20th century to preserve their palaeontological heritage. However, scientific colonialism is still reflected in many publications describing fossil specimens recovered from these countries. Here, we present examples of ‘palaeontological colonialism’ from publications on Jurassic-Cretaceous fossils from NE Brazil and NE Mexico spanning the last three decades. Common issues documented in these publications are the absence of both fieldwork and export permit declarations and the lack of local experts among authorships. In Mexico, access to many fossil specimens is restricted on account of these specimens being housed in private collections, whereas a high number of studies on Brazilian fossils are based on specimens illegally repositied in foreign collections, particularly in Germany and Japan. Finally, we outline and discuss the wider academic and social impacts of these research practices, and propose exhaustive recommendations to scientists, journals, museums, research institutions, and government and funding agencies in order to overcome these practices.

Keywords: scientific colonialism, parachute science, research ethics, palaeontological heritage, illegal fossil trade, Latin America

1. Introduction

Scientific advances played an important role in the expansion of European powers during colonial times. Scientific curiosity is noted as one of the key motivations behind the expeditions that led to the colonisation and annexation of regions in Asia, Africa, and the Americas [1]. As a result, many “exotic” specimens collected by naturalists or geologists from colonies were sent back to the respective colonial state, to adorn houses of high-ranked members of society or to be repositied in national scientific societies or institutions, with the purpose of scientific inquiry [1,2]. The latter led to the establishment of large museums to house the vast collections of curiosities brought back to Europe from overseas expeditions as well as imperial conquests within Europe. Although colonialism is frequently described in political, social, and military contexts, it is also present in many scientific practices still in use today. The institutionalisation of scientific disciplines, educational programmes, and academic organisations were all products designed to benefit colonial advancement [3].

This structure of colonial science — derived from the practice of science in the colonies — has given rise to ‘scientific colonialism’ in the postcolonial world, some of whose extractive aspects are sometimes referred to as parachute science [4–7], helicopter research [8,9] or even parasitic science [7]. Within scientific colonialism, middle and low income countries are perceived as suppliers of data and specimens for the high income ones, the contributions of local collaborators are devalued or omitted, and the legal frameworks in lower income countries are trivialised or even neglected [6,10–13]. In turn, colonist nations owe their wealth to the extractive colonial practices they have been carrying out for centuries, allowing them to accumulate knowledge, power and financial resources to continue these practices into the present day. These practices are also prevalent in the field of palaeontology [13].

As a reaction to a long history of colonial science practices, many countries, most notably several Latin American countries, adopted protective laws and regulations during the 20th century in order to preserve their biological, archaeological, and palaeontological heritage. Under these laws, fossils are considered to be property of the nation state, and their sale, purchase, and permanent exportation are prohibited [14–16]. Countries like Brazil [17], Argentina [14], Colombia [18] and Chile [19] also make it compulsory for a foreign party to be associated with a local institution in order to conduct fieldwork in the respective country and collect fossil samples. Brazil and Mexico prohibit the commercial trade of their fossils, require permits for their temporary export and, in the case of Brazil, the permanent export of specimens used for describing new species is not allowed [20,21]. These two countries, despite the presence of laws and regulations, still fall victim to scientific colonialism, including the illicit trafficking of fossil specimens. In fact, the illegal trade of fossils in Brazil has been blamed on the presence of laws; Martin [22,23] states that the very fact

that laws exist for the protection of these fossils could be the reason that officials can be bribed and
these fossils can be sold for a considerable price on the black market.

Both Brazil and Mexico are former European colonies with vast territories, large
sedimentary basins, and a huge palaeontological potential that remains relatively unexplored. These
characteristics, together with the predominance of an overall low income population and a local
currency devaluation, make them attractive targets for palaeontological colonialism. In the last
decades, the Crato and Romualdo formations of the Araripe Basin in NE Brazil and the Sabinas,
Parras and La Popa basins (Mexican Gulf) in NE Mexico have produced an unprecedented wealth
of Jurassic to Cretaceous (200–66 million years ago) fossils. These extremely rich fossil and well-
preserved deposits, known as *Lagerstätten*, have enriched our view of evolution revealing a
plethora of new vertebrates (figures 1 and 2), invertebrates, plants (figure 3) and fungi [24–27].
These exposures yield tantalizing examples of fossil preservation, including several soft-tissue
instances [28–32]. Most of the published research output on fossils from these regions, however,
has been led by foreign palaeontologists with few collaborations with local researchers. Many of
these studies are based on fossils that have been unethically and/or irregularly acquired and/or
exported [33]. Several published fossils lack contextual geographic and geologic information,
whereas many important specimens are in private or foreign collections, where they can be difficult
to access. Recent papers describing new fossil species such as the snake-like reptile *Tetrapodophis*
*amplectus* [34] (figure 1a) and the dinosaur “*Ubirajara jubatus*” [35] (figure 1c) both from the
Crato Formation (Araripe Basin) Brazil, as well as the plesiosaur *Mauriciosaurus fernandezi* [30]
(figure 2e) and the shark *Aquilolamna milarcae* [36] (figure 2f) from the Agua Nueva Formation
(Sabinas Basin), Mexico have raised a number of questions involving ethics, legal issues, and
science reproducibility. In this study, we present and discuss the academic and social impact of the
research published during 1990-2021 that likely represent examples of scientific colonialism in
Brazil and Mexico (Box 1). We chose this time period as (1) this is when ~~the~~ most research has
been carried out in the two basins that we chose as case studies and (2) many of the relevant laws
and regulations were established just before or in 1990. We also propose alternatives and
recommendations to scientists, journals, research institutions and government agencies in order to
overcome these practices.

2. Legal framework

2.1. Brazil

In Brazil, fossils are protected by Decree 4.146, published in 1942 [37] which states that fossils
cannot be privately owned and belong to the Union, and that fossil collecting requires an
authorization from Agência Nacional de Mineração, ANM (National Mining Agency, formerly the

National Department of Mineral Production). In 1990, the Brazilian government published Decree
98.830 (Box 1)[17] to regulate foreign scientific expeditions that collect biological or
palaeontological material (i.e. fossils) in the country. This law is regulated by Ordinance 55 from
Ministério de Ciência Tecnologia e Inovação, MCTI, formerly MCT [38] (see Box 2). According to
this legislation [38], any foreign party that wishes to permanently export specimens from Brazil
must have a permit from MCTI and a partnership with a Brazilian scientific institution (who will be
in charge of applying for the permit). Furthermore, Decree 98.830 explicitly states that fossil
holotypes, 30% of any collected taxon, and other specimens “whose permanence in the country is of
national interest” cannot be exported. A recent ordinance issued by ANM [39] reinforces the
necessity for foreign palaeontologists to comply with requirements stipulated by MCTI’s ordinance
of 1990. The original Brazilian laws and their English translations are available in Appendix A.

**2.2. Mexico**

A law regarding archaeological monuments was established in 1897 [40], as a reaction to the
looting of the Mayan site of Chitchén Itzá by Edward Thompson, then in charge of being the USA
consul in Mérida, Yucatán [41,42]. Currently, the law in force is The Federal Law of
Archaeological, Artistic and Historic Monuments and Zones, published in 1972 [43]. The Instituto
Nacional de Antropología e Historia, INAH (National Institute of Anthropology and History), was
created through an organic law in 1939 [41,44] in order to protect this heritage. A presidential
decree issued in 1986 [45], added the article 28bis to the law from 1972, and reformed the Organic
Law of INAH, making it responsible for overseeing any activities involving the discovery and
treatment of any fossil material, the delimitation of the boundaries of a fossiliferous site, and the
safeguarding of the material in a collection [16,46]. The law declares that fossils are property of the
Mexican Federation even if they are under custody of a private person (Articles 27, 28 and 28bis
[43]. Private collections, i.e. fossil collections owned by private individuals or companies, must be
registered by INAH [47]. Fossils in private collections are inalienable and imprescriptible, i.e. once
registered, they cannot be transferred to other collections [43]. Since 1986, Mexican law explicitly
forbids fossils to be commercially traded in Mexico [45,48]. In 1994, INAH created the National
Council of Paleontology to form a multidisciplinary and inter-institutional group with the aim of
reaching an agreement on what, how, and why to legislate the research on the palaeontological
heritage in Mexico [21,49]. Mexican law does not formally require foreign palaeontologists and
institutions to work with a local partner. However, the National Council of Paleontology recently
released a series of recommendations for palaeontological studies in the country [50] advising
foreign parties that wish to work in the country to previously notify the council (Box 2). The
original Mexican laws and their English translations are available in Appendix B.

2.3. UNESCO Convention on Illicit Trafficking

The UNESCO 1970 Convention on the “Means of Prohibiting and Preventing the Illicit Import, Export and Transfer of Ownership of Cultural Property” [51] was signed in November 1970 and came into effect in April 1972 as a response to the growth in the black market of cultural property since the 1950s. The 1970 Convention is in theory central to preventing the illicit trafficking of cultural property in the first place by promoting international cooperation between countries as a means to protect cultural heritage properly. Signatories of the convention thus acknowledge that the “illicit import, export and transfer of ownership of cultural property is one of the main causes of the impoverishment of the cultural heritage of the countries of origin of such property” and as such, the import, export, and transfer of ownership is prohibited under the convention. Thus, the signatories of the Convention are not only required to set up national laws and services for the protection of the cultural heritage but also to take the appropriate measures so that museums and other institutions within their territories are prevented from acquiring illegally exported cultural property from another country as well as to cooperate with regards to restitution of the object(s). This convention was ratified by several nation states including Brazil (in 1973) and Mexico (in 1972), both of whom consider palaeontological objects or sites as part of the cultural property, as well as Japan (in 2002) which includes geological features among other things and Germany (in 2007), which only considers archaeological objects as such.

3. Case studies

3.1. Case study 1: Palaeontology in the Sabinas, La Popa and Parras basins of Mexico

In northwestern Mexico there are several fossil sites with great palaeontological interest: Parras, La Popa, Rincon Colorado, Múzquiz and Vallecillo (figure 3*a, b*). They are included within the Sabinas, Parras and La Popa basins, that extend from the center of Coahuila to the southeast of Nuevo León [52]. Most of these sites are popularly known for the presence of dinosaur fossils, but also contain fossils of invertebrates, plants and several aquatic vertebrates (figure 2). The Muzquiz and Vallecillo deposits support the economic activities of the local community, where flagstone material from these deposits is commercially exploited for construction. During the extraction of the slabs, fossils of exceptional preservation can often be found. Some of these may be destroyed in the process of extracting flagstone slabs and others are kept by quarry workers [53]. Unfortunately, some specimens end up on the black market.

One of the most important sites considered in this study is the Vallecillo quarry in Nuevo León, which was first discovered and exploited for its lead, zinc and silver outcrops. Currently,

construction materials are extracted from the quarry. In the 1980s, inhabitants of the Vallecillo
village started to find several well-preserved vertebrate and invertebrate fossils from quarries [54].
The palaeontological value of the locality was recognised as early as 1997 [55]. Most of the
research on fossils from this locality has been intensively carried out by foreign researchers.
Material from the quarry was first exported in 1999-2000 to Germany, to the University of
Karlsruhe, where it remained until it was returned to Nuevo León in 2007 and deposited in the
Museo La Plomada [56].

We found 128 papers on Jurassic-Cretaceous macrofossils (excluding plants) from Sabinas-
La Popa-Parras basins and other sites in Coahuila and Nuevo León states in NE Mexico, published
between 1990-2021 (see Box 3, and electronic supplementary material, Table S1). Foreign
researchers led 46.88% of the publications (60 papers) (figure 5). The majority of this output, or
51.67% (31 papers), do not include local authors (i.e. an author based at a Mexican
scientific/academic institution based on the affiliations of the publications). In four papers [36,57–
59], the local authors included within the author list are not affiliated with any scientific institution.
Five of these papers are based on fossils stored in private collections (i.e. not in a recognised
museum or other official scientific repository), two of which describe new genera and species
[30,36]. Finally, two papers mention the purchase of a fossil [29,36] whereas another mentions that
a specimen “was obtained by a private collector from a local quarry worker” [60]. We found no
record of Mexican fossils being stored in foreign collections, except for a paper published in 1990
[61] that studied a collection of fossils claimed to have been collected in the 1930s.

[revised manuscript text omitted]

59 We found 71 papers on Cretaceous macrofossils (excluding invertebrates and non-holotype
60 material) from the Araripe Basin, published between 1990-2021 (see Box 3 and electronic

supplementary material, Table S2 for list of papers and methods used). The majority of these
publications (59.15%), were led by foreign researchers, and over half of them (57.14%) excluded
local Brazilian researchers (figure 5). A large proportion (88%) of the fossils described in these
publications (all holotype specimens) were taken from Brazil to be housed in foreign museum
collections and have not been returned. Of the publications describing fossils that were permanently
taken to foreign collections, not a single one reported exportation permits. Only one of these [80]
reported that the specimens were collected during fieldwork conducted by the authors, yet did not
mention a collecting permit, which would have been required by law. Several papers provide only
vague statements of provenance (e.g. “Araripe Basin, Brazil” [81], “Araripe Plateau, Brazil” [82]),
and do not mention fieldwork nor explain how the fossils ended in foreign collections. This fact,
together with the absence of reported exportation permits, leads us to consider that these fossils may
have been purchased (figure 5). Some publications inform that the fossils were “obtained from a
quarry workman” [83] or “from a fossil digger” [84] and in eight papers [85–92], the fossil
purchase is admitted by the authors.

Brazilian researchers, on the other hand, are responsible for 40.85% of the published
research (29 papers) on new species of Araripe vertebrates and plants during the same period
(figure 5). Three of these papers are based on fossils deposited in foreign collections and did not
mention exportation permits; one of these fossils is mentioned to have been collected in the 1960s
[93] whereas the two others [94,95] lack any data on provenance and probably were purchased by
the museums.

Two recent studies on particularly high-profile fossils, deserve our attention here due to the
questionable practices exposed therein. The 2015 publication of the putative legged-snake
*Tetrapodophis amplexus* from Araripe in the journal *Science* by Martill et al. [34] (figure 1a)
caused quite some controversy [96]. To start with, the publication did not involve any Brazilian
researchers or institutions. The authors also claimed that the specimen was permanently available in
a museum. The fossil, however, belongs to a private collector in Germany [96–98] and access to
this specimen has reportedly been made difficult for other researchers who wish to study it [96,97].
Finally, the authors did not provide any evidence that the fossil was legally collected and exported
from Brazil.

More recently in 2020, the collection, study, and publication describing a new dinosaur
species “*Ubirajara jubatus*” (figure 1c), also from the Araripe Basin, similarly did not include
Brazilian researchers or institutions. This paper was published in the journal *Cretaceous Research*
but has been temporarily retracted at the time of our study while an investigation is carried out [35].
The now-retracted publication mentioned that the fossil material was collected and removed from
Brazil after 1990, which attracted much attention from the media. Through a journalistic report

[99], the authors presented a signed document from 1996 by an agent at the regional office of the National Department of Mineral Production named José Betimar Melo Filgueira. The document in question is surprisingly vague, without specifying how many or which kind of fossils were exported, mentioning only "two boxes with limestone samples containing fossils". The document also does not identify the applicant, nor the destination of the material. There was also no information on whether this was a temporary or permanent exportation, and no mention of any collaborating Brazilian institution. The aforementioned agent has previously collaborated with one of the authors [100] of the "*Ubirajara jubatus*" paper around the same time that the specimen was exported from Brazil, which may represent a conflict of interest. Furthermore, no mention is made by the authors of the study about having the necessary export authorization from MCTI as per Decree 98.830 [17]. The journal *Cretaceous Research* has become a popular venue for hosting many studies on ethically questionable fossils (see section 9.2 below).

Several of the above mentioned studies led by foreign researchers represent clear cases of scientific colonialism. As seen in the previous case study in Mexico, the examples here also show a general aim to avoid interaction with local scientists, but to an even greater extent than seen in Mexico. To make matters worse, the overwhelming majority of these studies are based on fossils likely to be both illegally purchased and exported. Our survey shows that the primary destinations of most of the illicitly exported fossils are the Museum für Naturkunde Berlin (13 plant holotypes) and Staatliches Museum für Naturkunde Karlsruhe (10 vertebrate holotypes) (see electronic supplementary material, Table S2). Because our study was limited to fossil holotypes, and we did not include arthropods and other invertebrates, the above numbers represent only a small portion of the fossils that have been irregularly taken from Brazil to foreign institutions. A preliminary search for Araripe arthropod papers on Google Scholar (see electronic supplementary material, Table S3), however, indicates that at least 47 holotypes of insects, arachnids and chilopods were illegally transferred to Staatliches Museum für Naturkunde Stuttgart, at least seven to Senckenberg Museum in Frankfurt am Main, four to Museum für Naturkunde Berlin, four to Staatliches Museum für Naturkunde Karlsruhe, and several others to various institutions and private collections, totaling a minimum of 90 Araripe holotypes in German collections. In addition, our survey of papers located at least four vertebrate holotype fossils housed in Japanese collections, along with seven more arthropod holotypes identified through a preliminary search on Google Scholar.

4. Beyond the Araripe and Sabinas basins

Colonial palaeontology practices in Brazil and Mexico are not limited to Sabinas-La Popa-Parras and Araripe basins. In México, important Pleistocene mammal specimens from the Yucatán Peninsula have been targeted as well [101–106]. A 2012 study found that foreign-led research has

[revised manuscript text omitted]

nationalists, and we are not aware of major engagement by foreign research communities with
groups attempting to circumvent these protections. We therefore must conclude that the existence of
heritage laws is not an obstacle to good palaeontological research [177].

However, it can be difficult for foreign researchers to navigate local legislation and
bureaucracy, and cooperation with local institutions in Brazil and Mexico is essential in this regard
(and required in the case of the former). In addition to assisting with legal procedures, the institution
can properly guide foreign researchers on specific legislation regarding the collection, study, and/or
temporary export of fossils. The relevant laws that apply to palaeontological work in Brazil and
Mexico have been provided in full and translated to English herein (see Appendix ~~1~~ and ~~2~~). The
bureaucracy that may annoy and frustrate some helps to regularise and supervise, so that the local
legislation is complied with and the local heritage is protected.

5.7. Commercial exploitation of fossils helps science [178,179].

This assumption suggests that through commercial exploitation of fossil deposits (e.g. mining, quarrying) and the trading of extracted fossils may result in more fossils ultimately being uncovered. However, a greater number of fossils does not necessarily imply a benefit for science. When fossils are openly traded, the exploitation of fossil deposits can become uncontrolled, likely resulting in a loss of important provenance information about that material. The removal of fossils without any documentation of geological information reduces the scientific value of these specimens; new fossil material may lead to the discovery of new species but, without context, these specimens cannot inform on the ecology or evolution of these organisms. In the case of permissible trade (i.e. allowing only the sale of certain types of fossils and/or fossils from certain locations), as in Morocco, the law of “supply and demand” must be considered. The commercialization of this type of rare object (1) leads to an increase in inequality in science, by concentrating this type of material in institutions and countries with the financial means to acquire them, (2) encourages the existence of private collections, which can be an obstacle to the reproducibility of science, and (3) stimulates the artificial modification of fossils [180,181] to obtain a better market price. The latter can be exemplified by a case from the Araripe Basin itself. Martill et al. [85] studied an illicitly acquired, artificially “enhanced” specimen from Brazil and only discovered these heavy modifications during the course of their work. The case left the authors so “irritated” that they decided to express this frustration in the name of the new taxon: *Irritator challengerii*.

Commercial exploitation can be an ally in specific cases when it involves controlled, regulated, and documented collection of material. Nonetheless, this discussion should be led by the local community, together with effective communication initiatives from palaeontologists and local experts to highlight the scientific and cultural importance of fossils.

6. Implications for science and the local community

6.1. Private collections can interfere with the reproducibility of science and prevent access to information to the scientists and the general public.

Fossil specimens and the data obtained from them need to be accessible by scientists, not only for reproducibility and replicability, but also for verification and comparison purposes. A private collection is usually one that is in the ownership of a personal or corporate entity and any access to the material is at the discretion of the owner [182]. However unless the appropriate arrangements are made, there is a risk of losing these fossil materials due to changing circumstances, e.g. death or illness, or changes in personal finances that necessitate selling off parts of a collection. Current and future research relies on the permanent accessibility and stable storage of these materials, which

unless placed in a public trust, is rarely the case for privately owned collections [120,183]. Private
collections also hinder access of information to the general public. Science is a public endeavour,
often funded by tax money and as such, scientists have a responsibility to relay the findings and
provide access to materials - through museums - to the public. When fossils are stored in private
collections, this negates the public's ability to not only enjoy but also scrutinise scientific research
arising from public funds.

**6.2. The purchase of fossils, besides being currently illegal in these countries, does** 15 **not benefit the local community in the long term.**

Fossil deposits are finite. With the depletion of the resource, the living wage earned from any trade
quickly becomes inaccessible to people financially dependent on it. In addition, the impacts of
mining activity, including environmental damages and tailing monitoring in many cases, will be
borne by the community long after the resource is exhausted. It is a trade-off whose cost generally
falls on the most vulnerable citizens in the long run.

[revised manuscript text omitted]

**6.4. It affects ethical partnerships between local and foreign parties**

The *modus operandi* of scientific colonialism, in palaeontology and beyond, generally negatively
affects the local image of foreign researchers that wish to cooperate with local palaeontologists
through an equitable partnership. Parachute science practices generate distrust towards foreign
parties as a whole, regardless of which researcher is the leader of a group or which institution the
researcher(s) represents. Conversely, local researchers who wish to partner with foreign colleagues
are often looked upon with suspicion by other local researchers due to the bad image generated by
third party colonial practices. Overall this situation prevents the progress of international scientific
cooperation and hampers the development of local science and local researchers.

6.5. Negative impact on local science development by making it difficult to access specimens.

Allowing fossil materials to be repositied in foreign institutions restricts the advancement of the palaeontological discipline in their countries of origin. When fossils are taken away to 
[revised manuscript text omitted]
 [224]. The circumstances of this case draws parallels with the *Tarbosaurus* one. After being alerted by palaeontologists on Facebook to the online auction of one of the biggest and almost complete *Anhanguera santanae* specimens, Brazilian palaeontologists alerted the Brazilian Public Prosecutor’s Office which immediately launched an investigation with the help of French authorities [78,224]. Another case of successful repatriation is that of Chinese fossils from Australia in 2008 thanks to the efforts of Australian palaeontologist John Long with the help of his Chinese collaborators who worked together with the Australian Federal Police following

[revised manuscript text omitted]

Other paleontological journals apparently have strict policies regarding ethics and legality
but unfortunately do not put them into practice. *Cretaceous Research*, for example, states in its
guide for authors that “Fossil material of uncertain or dubious provenance will not be accepted for
publication in *Cretaceous Research*. This includes material currently housed in museum collections
which lack detailed field collecting records, and/or which provenance cannot be definitively
ascertained with certainty” [241]. This journal, however, notoriously publishes to this day numerous
fossils of highly questionable origin from Araripe [242–245] (including “*Ubirajara jubatus*” [35],
temporarily retracted after complains from other researchers and social media pressure) and even
amber from Myanmar (19 papers in 2021 so far), the latter already being formally banned by a

number of palaeontology journals for ethical and legal reasons [120–122] (see discussion in Section
4).

Both in Brazil and in Mexico, fossil trade or export without a permit is illegal. Manuscripts
involving fossils from these countries that only include vague statements involving the acquisition
of a specimen or that cannot provide full information regarding how the fossil was obtained, should
not be considered for publication. Brazilian fossils collected after 1942 and stored in foreign
collections should be regarded with high suspicion. Authors that claim that a Brazilian fossil in a
foreign collection was obtained before 1942 should be able to produce evidence of this. We are
aware that Brazil's ANM authorized the legal exportation of some Brazilian fossils after this date
(e.g. *Prionosuchus plummeri* specimens stored at the Natural History Museum in London, UK,
[246]), but these are very rare cases that involve non-type specimens, and authors that study this kind
of material should demonstrate the pertinent documentation from the keeper institutions. Any
Brazilian holotype collected after 1990 that is stored in a foreign collection represents a violation of
Brazilian laws as demonstrated above. Authors must obtain and provide proper documentation from
ANM and MCTI that demonstrates that these fossils were collected and exported legally. Fossils
that are kept in private collections and not in research institutions, should also not be considered for
publication, especially when involving new taxa. Mexican fossils having an INAH registration
number are not necessarily repositied in museums or universities because this government agency
also registers private collections. Moreover, in order to encourage registration, INAH does not ask
questions regarding provenance or authenticity of the fossils. Authors that wish to publish Mexican
fossils registered by INAH should provide documentation by this organ proving that the fossil in
question is available to scientists in a research institution and not part of a private collection. As
mentioned above, some of the foreign researchers that usually work on these fossils openly
advocate breaking local laws and regulations [10,11,168]. The burden of proof is on the side of the
authors of the manuscripts. We strongly recommend that editors refuse to publish studies on these
fossils unless their legal status has been clearly proven with supporting documentation by the
authors. We also recommend that reviewers go beyond the requirements of the journal to ask these
questions and demand documentation proactively, in the cases where journal editorial policies are
lagging or inadequate. While it is imperative that research complies with the national legislature,
there is no legal requirement to engage in work that is ethical or abstain from parachute science. We
therefore must also recommend that editors, reviewers, and authors be mindful of the implications
of research that is not equitable.

9.3. Recommendations to local governments and authorities

Governments should review laws regularly to avoid creating legal loopholes by consulting experts, ensuring that no conflict of interest exists, and enforce current laws and regulations. In August 2020, a bill was passed in the Nuevo León Congress that redefined the material from Vallecillo (in the Sabinas Basin), collected for over two decades, as “unusual engravings in carbonate limestone”. One of the main concerns, as reported by journalists at the time [247] and voiced by the Council of Palaeontology of INAH [248], is that the new law opens the possibility to openly trade fossils extracted from the Vallecillo quarry thus bypassing fossil protection legislations [247,248]. In order to make our study more accessible to local authorities and policy makers, we have made translations to Spanish and Portuguese available (see supplementary electronic material Translation S1 and S2).

However, due to the asymmetry of power between the governments of formerly colonized countries and major colonial powers of the Global North, the burden to prevent smuggling of fossils poached in countries with robust heritage laws must also be taken up by the countries where these fossils are ultimately purchased. Efforts to counter smuggling of cultural and national heritage objects as well as fossils and protected wildlife have been undertaken in some countries of the Global North, notably Canada and the US [219,249], but this effort has largely focused on high profile items (e.g. dinosaur skeletons) rather than general trade in contraband heritage objects. Other countries, such as Germany, have previously been referred to as being one of the centers of international trade of illicit antiquities [250]. More robust adherence to international conventions on smuggling of heritage objects by destination countries is a critical piece of any effort to bolster local fossil protection legislation.

9.4. Recommendations to research institutions, funding agencies, and reviewers

Palaeontology is a science that captures wide public interest [251,252]. Colonial science practices repeatedly carried out by some palaeontologists can evolve into a negative public perception towards an academic institution that tolerates them. Complying with local regulations is not only a logical step before undertaking an expedition or research on internationally-sourced specimens, but should also be an expected ethical practice for palaeontologists and scientists in general. Some professional palaeontological societies currently have a Code of Conduct addressing this specific point that their members must comply with [253,254]. Museums and universities should advise and support their palaeontologists that perform research in low-income or resource-poor countries in order to avoid conducting colonial science practices. Universities should also consider including science history and ethics courses as part of their undergraduate and graduate programmes to ensure that the future generation of palaeontologists receive adequate training on ethical issues within palaeontology. Funding agencies should make it compulsory for applicants to demonstrate that they

will comply with the laws of the countries they wish to carry out research in. Failure to do this
should result in the funding being terminated. Funders should also have a requirement that
equitable, institutionalized cooperation with local counterparts be facilitated in a way that can
benefit both parties involved in the collaboration. As with manuscript reviews, grant reviewers
should consider preemptively scrutinising proposed partnerships and international fieldwork
irrespective of whether the funder currently requires this oversight. Reviewers should also be
mindful of the ethical implications of proposed partnerships and outcomes, outside of legal
requirements.

Museums share a burden of responsibility for colonial practices that is difficult to ignore
[255]. It is hard to imagine that nearly a hundred Araripe holotype fossils (and presumably an even
larger number of non-type fossils) made it illegally to museums in Germany without the knowledge
or even support from their respective curators. As seen above, there are several cases in which
authors openly admit that fossils were purchased. Details of purchases are even recorded on
museum labels within collections (figure 10a) and there are even cases in which Araripe fossils are
being sold by the museum gift shops (figure 10b). Museums should adopt strict policies regarding
the reception of specimens in their collections. Banning the admission of fossils with dubious
provenance data or from countries that prohibit their exportation, as is the case of Brazil and
Mexico, should be embraced as a formal policy.

**10. Conclusions**

We recommend that museums, universities, and funding agencies avoid facilitating research or
researchers that are involved in colonial scientific practices, especially when there are signs of
violation of local laws and regulations, such as the illegal purchase and export of fossils. It is
equally important that journals make it mandatory for authors to provide applicable research and
exportation permits, and refuse to publish research that is produced through unethical and irregular
activities, such as the cases enumerated herein. Foreign researchers must respect local laws and
regulations and engage in constructive, ethical, and equitable partnerships. The extractive history of
colonialist palaeontology cannot be rewritten, but we can forge a new path based on respectful
cooperation that mutually benefits both foreign and local institutions, as well as the local
communities that remain the custodians of their palaeontological heritage. Colonialist science must
end, not only because it is a destructive, pervasive practice that has deep, negative ethical and social
implications, but also because it often produces dubious, non-reproducible research.

Acknowledgements

This manuscript had input from other colleagues that preferred to remain anonymous. We appreciate the help provided by Alberto Blanco Piñón through our discussions on the Sabinas fossils as well as the photographs he provided. We are thankful to Felisa Aguilar Arellano (Consejo de Paleontología - INAH) for her valuable help with Mexican fossil laws. Chico Camargo supported us in extracting the Twitter data in this study. Our thanks also go to: Paolo Schirolli (Museo Civico di Scienze Naturali di Brescia) and Lorenzo Marchetti (Museum für Naturkunde Berlin) who provided information regarding Italian laws, Jeff Liston (Royal Tyrrell Museum of Palaeontology) and John Long (Flinders University) for providing information on previous repatriations of fossils, and Edenilce P. Batista (Universidade Regional do Cariri) for information on Araripe plants. The photograph of *Tetrapodophis amplectus* (figure 1f) was provided by Michael Caldwell (University of Alberta), images in figure 2e-f are courtesy of El Norte - Grupo REFORMA, photograph of La Mula Quarry (figure 3b) was provided by Selene Velázquez, and the art in figure 8d was created by Saulo Daniel Ferreira Pontes. The manuscript also greatly benefited from comments by Sarah Greene (University of Birmingham) and Shazia Kurmoo (Ministry of Foreign Affairs, Mauritius). We are also grateful to Alexandra Elbakyan and the Sci-Hub project for providing access to several papers that were necessary to this study. NBR was supported by Deutsche Forschungsgemeinschaft (KI 806/17-1). The funders had no role in the study design, data collection and analysis, decision to publish, or preparation of the manuscript.

References

1. Adas M. 2008 Colonialism and Science. In *Encyclopaedia of the History of Science, Technology, and Medicine in Non-Western Cultures* (ed H Selin), pp. 604–609. Dordrecht: Springer Netherlands. (doi:10.1007/978-1-4020-4425-0_8518)
2. Aldrich R. 2009 Colonial Museums in a Postcolonial Europe. *African and Black Diaspora: An International Journal* **2**, 137–156. (doi:10.1080/17528630902981118)
3. Andersen C, Clopot C, Ifversen J. 2020 Heritage and interculturality in EU science diplomacy. *Humanities and Social Sciences Communications* **7**, 1–8. (doi:10.1057/s41599-020-00668-8)
4. de Vos A. 2020 The Problem of ‘Colonial Science’. *Scientific American*, 1 July. See <https://www.scientificamerican.com/article/the-problem-of-colonial-science/>.
5. North MA, Hastie WW, Hoyer L. 2020 Out of Africa: The underrepresentation of African authors in high-impact geoscience literature. *Earth-Science Reviews* **208**, 103262. (doi:10.1016/j.earscirev.2020.103262)
6. Stefanoudis PV, Licuanan WY, Morrison TH, Talma S, Veitayaki J, Woodall LC. 2021 Turning the tide of parachute science. *Current Biology* **31**, R184–R185. (doi:10.1016/j.cub.2021.01.029)
7. The Lancet Global Health. 2018 Closing the door on parachutes and parasites. *The Lancet Global Health* **6**, e593. (doi:10.1016/S2214-109X(18)30239-0)
8. van Groenigen JW, Stoof CR. 2020 Helicopter research in soil science: A discussion. *Geoderma* **373**, 114418. (doi:10.1016/j.geoderma.2020.114418)
9. Minasny B, Fiantis D, Mulyanto B, Sulaeman Y, Widyatmanti W. 2020 Global soil science

research collaboration in the 21st century: Time to end helicopter research. *Geoderma* **373**, 114299. (doi:10.1016/j.geoderma.2020.114299)

10. Martill DM. 2011 Protect - and die. *Geoscientist Online November-2011*. See <https://www.geolsoc.org.uk/Geoscientist/Archive/November-2011/Protect-and-die>.
11. Martill DM. 2018 Why palaeontologists must break the law: a polemic from an apologist. *The Geological Curator* **10**, 641–649.
12. Albersdörfer R. 2018 Fossil legislation - protection or destruction? *The Geological Curator* **10**, 603–605.
13. Raja NB, Dunne E, Khan TM, Nätscher P. 2020 The Overlooked Realities of Sampling Bias in the Fossil Record. p. 356351. (doi:10.1130/abs/2020AM-356351)
14. Fernández D, Luci L, Cataldo C, Pérez D. 2014 Paleontology in Argentina: history, heritage, funding, and education from a southern perspective. *Palaeontologia Electronica* (doi:10.26879/146)
15. da Conceição DM, Tavares TMV, Cisneros JC, Kurzawe F, de Alcântara Alencar M, Filho MP, da Silva-Melo A, Aires HA. 2020 Geoconservation of Permian Petrified Forests from Northeastern Brazil. In *Brazilian Paleofloras* (eds R Iannuzzi, R Rößler, L Kunzmann), pp. 1–36. Cham: Springer International Publishing. (doi:10.1007/978-3-319-90913-4_13-1)
16. Guerrero-Arenas R, Arellano FJA, Mendoza LA, Jiménez-Hidalgo E. 2020 How is the paleontological heritage of Mexico and other Latin American countries protected? *Paleontología Mexicana* **9**, 83–90.
17. Brazil. 1990 Decreto No 98.830, de 15 de Janeiro de 1990.
18. Ministerio de Minas y Energía. 2018 Decreto 1353 del 31 de Julio de 2018.
19. Ministerio de Educación. 1990 Decreto Supremo N° 484, de 1990, del Ministerio de Educación: Reglamento Sobre Excavaciones Y/O Prospecciones Arqueológicas, Antropológicas Y Paleontológicas.
20. Viana MSS, Carvalho I de S. 2019 *Patrimônio Paleontológico*. Editora Interciência. Editora Interciência: Rio de Janeiro.

[revised manuscript text omitted]

*AAPG Bulletin* **55**, 2130–2140.
- 208. Ayala-Castañares A, Knox RA. 2000 Opportunities and challenges for Mexico-US
cooperation in ocean sciences. *Oceanography* **13**, 79–82. (doi:10.5670/oceanog.2000.15)
- 209. Kurzawe F, Iannuzzi R, Merlotti S, Röbller R, Noll R. 2013 New gymnospermous woods from
the Permian of the Parnaíba Basin, Northeastern Brazil, Part I: *Ductoabietoxylon*,
*Scleroabietoxylon* and *Parnaiboxylon*. *Review of Palaeobotany and Palynology* **195**, 37–49.
(doi:10.1016/j.revpalbo.2012.12.004)
- 210. Kurzawe F, Iannuzzi R, Merlotti S, Rohn R. 2013 New gymnospermous woods from the
Permian of the Parnaíba Basin, Northeastern Brazil, Part II: *Damudoxylon*, *Kaokoxylon* and
*Taeniopitys*. *Review of Palaeobotany and Palynology* **195**, 50–64.
(doi:10.1016/j.revpalbo.2012.12.005)
- 211. Neregato R, Röbller R, Rohn R, Noll R. 2015 New petrified calamitaleans from the Permian of
the Parnaíba Basin, central-north Brazil. Part I. *Review of Palaeobotany and Palynology* **215**,
23–45. (doi:10.1016/j.revpalbo.2014.12.006)
- 212. Neregato R, Röbller R, Iannuzzi R, Noll R, Rohn R. 2017 New petrified calamitaleans from the
Permian of the Parnaíba Basin, central-north Brazil, part II, and phytogeographic implications
for late Paleozoic floras. *Review of Palaeobotany and Palynology* **237**, 37–61.
(doi:10.1016/j.revpalbo.2016.11.001)
- 213. Neregato R, Röbller R, Noll R. 2020 Growth Architecture Diversity Among Permian
Calamitaleans in Brazil. In *Brazilian Paleofloras* (eds R Iannuzzi, R Röbller, L Kunzmann),
pp. 1–32. Cham: Springer International Publishing. (doi:10.1007/978-3-319-90913-4_8-1)
- 214. Colwell C. 2015 Curating Secrets. *Current Anthropology* **56**, S263–S275.
(doi:10.1086/683429)
- 215. Breske A. 2018 Politics of Repatriation: Formalizing Indigenous Repatriation Policy. *Int J*
*Cult Prop* **25**, 347–373. (doi:10.1017/S0940739118000206)
- 216. Peirson-Hagger E. 2019 Can we decolonise the British Museum? *News Statesman*, 20 July.
See [https://www.newstatesman.com/culture/art-design/2019/07/can-we-decolonise-british-](https://www.newstatesman.com/culture/art-design/2019/07/can-we-decolonise-british-museum)
[museum](https://www.newstatesman.com/culture/art-design/2019/07/can-we-decolonise-british-museum).
- 217. Vogel G. 2019 Countries demand their fossils back, forcing natural history museums to
confront their past. *Science* (doi:10.1126/science.aax4867)
- 218. Hicks D. 2020 *The british museums: the Benin Bronzes, colonial violence and cultural*
*restitution*. London: Pluto Press.
- 219. U.S. Attorney’s Office. 2014 Manhattan U.S. Attorney Announces Return To Mongolia Of
Fossils Of Over 18 Dinosaur Skeletons. See [https://www.justice.gov/usao-sdny/pr/manhattan-](https://www.justice.gov/usao-sdny/pr/manhattan-us-attorney-announces-return-mongolia-fossils-over-18-dinosaur-skeletons)
[us-attorney-announces-return-mongolia-fossils-over-18-dinosaur-skeletons](https://www.justice.gov/usao-sdny/pr/manhattan-us-attorney-announces-return-mongolia-fossils-over-18-dinosaur-skeletons) (accessed on 6
April 2021).
- 220. Kelley N. 2012 Stop the auction of illegally collected Mongolian dinosaur fossils. *Change.org*.
See [https://www.change.org/p/heritage-auctions-stop-the-auction-of-illegally-collected-](https://www.change.org/p/heritage-auctions-stop-the-auction-of-illegally-collected-mongolian-dinosaur-fossils)
[mongolian-dinosaur-fossils](https://www.change.org/p/heritage-auctions-stop-the-auction-of-illegally-collected-mongolian-dinosaur-fossils) (accessed on 6 April 2021).
- 221. Society of Vertebrate Paleontology. 2013 Soc of Vert Paleontology members awarded by
Mongolia for role in repatriation of stolen fossils. *EurekaAlert!*. See
http://www.eurekaalert.org/pub_releases/2013-07/sovp-sov072613.php (accessed on 6 April
2021).
- 222. Minjin B. 2017 Fossil Repatriation – Institute for the Study of Mongolian Dinosaurs.
- 223. Xinhua. 2018 Mongolia to recover dinosaur fossils from South Korea, France. See
http://www.xinhuanet.com/english/2018-03/19/c_137050769.htm (accessed on 6 April 2021).
- 224. de Oliveira Andrade R. 2019 Brazil wins legal fight over 100-million-year-old fossil bounty.
*Nature* **570**, 147–147. (doi:10.1038/d41586-019-01781-8)
- 225. Long J. 2017 Dinosaur embryo returned to China, but many fossils fall victim to illegal trade
and poor protection. *The Conversation*, 21 May. See [http://theconversation.com/dinosaur-](http://theconversation.com/dinosaur-embryo-returned-to-china-but-many-fossils-fall-victim-to-illegal-trade-and-poor-protection-77349)
[embryo-returned-to-china-but-many-fossils-fall-victim-to-illegal-trade-and-poor-protection-](http://theconversation.com/dinosaur-embryo-returned-to-china-but-many-fossils-fall-victim-to-illegal-trade-and-poor-protection-77349)
[77349](http://theconversation.com/dinosaur-embryo-returned-to-china-but-many-fossils-fall-victim-to-illegal-trade-and-poor-protection-77349).

[revised manuscript text omitted]

Box 1. Clarification

All views expressed in this paper rely solely on information, or lack thereof, provided in the publications discussed herein. We do not assume that the authors of the papers here discussed have violated or intended to violate any local laws or regulations. Neither do we assume that all of the co-authors of a particular paper concur with irregular or unethical practices eventually made by another co-author or by an institution.

Box 2. Portions of Decree 98.830 from 1990 and Ordinance 55 from 14/03/1990 of the Minister of Science and Technology that are relevant to foreign palaeontologists.

Decree 98.830 from 1990

Article. 3 The activities referred to in Article I will only be authorized as long as there are co-participation and co-responsibility of a Brazilian institution with a recognized technical-scientific concept in the research field correlated with the work to be developed, according to the assessment of the National Council for Scientific and Technological Development (CNPq).

Ordinance 55 from 14/03/1990 of the Minister of Science and Technology: "Regulates the collection of scientific material by foreigners, according to Decree 98.830 / 1990"

42 - The MCT, through the co-participant and co-responsible Brazilian institution, will retain, from the collected material, for the destination to Brazilian scientific institutions, the following items:

(...)

e) all fossil type-specimens;

f) At least 30% of the specimens of each taxon identified at any time;

g) other specimens, data or materials, whose permanence in the country is of national interest.

Box 3. Recommendations for palaeontological studies in the country by National Council of Paleontology (INAH) [50]:

In the event of the intervention of an academic partnership from foreign institutions as co-responsible for the project, the Paleontology Council must be notified in a timely manner of their participation and the work they will carry out within the research project. The co-manager of the foreign institution must deliver in writing and with a handwritten signature an official letter in

which they undertake to send INAH a report of the results obtained from their participation, as well
as the products generated, once the project is concluded.

Electronic supplementary material:**Table S1.** Sabinas, La Popa and Parras basins fossil publications by foreign authors

DOI: 10.5061/dryad.8sf7m0cnd. Available at:

https://datadryad.org/stash/share/ebBibeh8gnm2lvrXMvZq07eB_5zzhA5JQ2OQnN500ts**Table S2.** Araripe fossil publications by foreign authors (vertebrates and plants)

DOI: 10.5061/dryad.g79cnp5q1. Available at:

<https://datadryad.org/stash/share/KJJvxFGJNSjl7rtGlzIR3CEeDzp4-pLEEKrRnez1798>**Table S3.** Preliminary list of Araripe fossil arthropod publications

DOI: 10.5061/dryad.mw6m905wv. Available at:

<https://datadryad.org/stash/share/PNYVwUjM8vFasi6jcLNI2T6bxlGYdCH0LxUAnEBsWH0>**Table S4.** List of palaeontology museums and postgraduate courses in Brazil with palaeontology advisors

DOI: 10.5061/dryad.pc866t1nz. Available at:

<https://datadryad.org/stash/share/g16fg3iQgQoC5dNCCGON6HM0uUL6dt7LOARhBTz7Z4I>**Table S5.** List of palaeontology museums and postgraduate courses in Mexico with palaeontology advisors

DOI: 10.5061/dryad.n8pk0p2vp. Available at:

<https://datadryad.org/stash/share/sGadbKQ2aQfF8VEC5QDSvMI9Bv2J7eeILl-VT-h47E>**Translation S1.** Complete article in Portuguese

Will be made available if approved by the journal.

Translation S2. Complete article in Spanish

Will be made available if approved by the journal.

Appendix A. Laws in Brazil (includes English translations)

DOI: 10.5061/dryad.cc2fqz664. Available at:

https://datadryad.org/stash/share/neDIPnhuj0WJc-4YyIm0gH--gO_uiKiaUjQwpCjHkbY**Appendix B.** Laws in Mexico (includes English translations)

DOI: 10.5061/dryad.d2547d82x. Available at:

https://datadryad.org/stash/share/eYJbLx9wNnQ6vCEvDgMKHv5dB9Yn_BAsXrzDHTFksQM

Figure 1. Holotype vertebrate fossils from Araripe Basin, Brazil, stored in foreign collections. (a) SMNK PAL 29241, proposed holotype skeleton of the feathered dinosaur "Ubirajara jubatus" [1], paper temporarily retracted by publisher at the time of preparing this study), (b) SMNK PAL 3828, holotype of the pterosaur *Ludodactylus sibbicki* [88]. (c) SMNK 2344 PAL holotype of the pterosaur *Tupandactylus navigans* [254], (d) SMNS 58022 holotype of *Irritator challengeri* [85] (e) SMNK PAL 3804, holotype of the crocodyliform *Susisuchus anatoceps* [253], (f) Private collection BMMS BK 2-2, holotype of the putative legged-snake *Tetrapodophis amplexus* [34], photograph by Michael Caldwell.

934x983mm (72 x 72 DPI)

Figure 2. Fossils from the Sabinas Basin stored in a private collection. (a) cf. *Tselfatia formosa*, ~ 750mm body length. (b) cf. *Belenostomus longirostris*, ~700mm body length. (c) pachyrhizodontid fish ~750mm body length. (d) chelonia cf. *Terlinguachelys* sp., ~300mm body length. (e) Holotype of plesiosaur *Mauriciosaurus fernandezii* (Frey et al. 2017). (f) Holotype of *Aquilolamna milarcae* [36]. All fossils are deposited in the collection registered by INAH as REG2544PF, which is housed by Mauricio Fernández (seen in the photograph) in Monterrey, Nuevo León, Mexico. (e, f) Image captures from video by Grupo Reforma Youtube Channel (El Norte - Grupo REFORMA 2021).

Figure 3. Fossil sites at the Sabinas, La Popa and Parras basins (NE Mexico) and Araripe Basin (NE Brazil). (a) La Mula Quarry, North of Múzquiz, Coahuila. (b) Vallecillo Quarry in Nuevo León State, with quarry worker Ramón Ramírez. (c) Nova Olinda Quarry in Ceará State. Photographs (a) courtesy of Alberto Blanco-Piñón, and (b) by Selene Velázquez.

402x328mm (118 x 118 DPI)

Figure 4. Publications on Jurassic and Cretaceous fossils from Sabinas, La Popa and Parras basins, and other sites in Coahuila and Nuevo León states between 1990-2021 (Plants and microfossils excluded). (a) Issues detected in the publications. (b) Current location of the fossils. See electronic supplementary material, Table S1, for list of publications and description of methods used.

Figure 5. Publications on Cretaceous fossils from Araripe Basins, Brazil, between 1990-2021(only holotypes, invertebrates excluded). See electronic supplementary material, table S2, for list of publications and description of methods used. (a, c) Current location of the fossils. (b) Issues detected in the publications.

Figure 6. Museums, natural monuments, and institutions providing postgraduate courses related to palaeontology in Mexico and Brazil. See detailed list in electronic supplementary material, tables S3 and S4.

Figure 7. Comparison of papers published by country during 2000–2018. Data from National Science Foundation from USA available through The World Bank at: <https://data.worldbank.org/indicator/IP.JRN.ARTC.SC>

582x395mm (72 x 72 DPI)

Figure 8. Outreach activities and public interest in palaeontology in Mexico and Brazil. (a) The cross of Picos de Pato (duck-bill dinosaurs) and Tiranosaurios streets at Rincón Colorado, Coahuila, México (with palaeontologist Giuseppe Leonardi). (b) School students learn how to find fossils in the Jovens Paleontólogos (Young Palaeontologists) Project in Nova Olinda, Ceará, Brazil, by Universidade Regional do Cariri (URCA). (c) Meeting of Paleontólogos Aficionados de Sabinas A.C. (Civil Association of Amateur Palaeontologists of Sabinas) in Coahuila (René Hernández Rivera and Jim Kirkland seen in the photograph). This association created the Museo Paleontológico de Múzquiz in 2005 [53]. (d) Fan art with #UbirajaraBelongstoBR hashtag posted on Twitter in December 2020 (credit: Saulo Daniel Ferreira Pontes, @saulodfp).

Figure 9. Posts on Twitter.com using the hashtag #UbirajaraBelongstoBR between 13 December 2020 and 31 March 2021.

Figure 10. (a) acquisition of Araripe fossils by Staatliche Museum für Naturkunde Stuttgart. SMNS 58022 holotype of the dinosaur *Irritator challengeri*, (label says "purchased from M. Kandler 1991"); SMNS 55414 pterosaur (indeterminate genus), (label says "purchased from K. H. Frickhinger, Planegg in Munich, 5.6.187" [sic]); 82001 pterosaur (indeterminate genus), (label says "purchased from K. H. Frickhinger Planegg in Munich, 5.6.1987, together with 55404-55415 for the price"); 56994 pterosaur *Tropeognathus robustus*, (label reads "acquired from C. Novaes Ferreira, São Paulo, Brazil (7.11.1990)". (b) Araripe fishes (*Dastilben* sp.) being sold at a souvenir shop in Staatliche Museum für Naturkunde Karlsruhe in 2011. Commerce and exportation of fossils has been forbidden in Brazil since 1942 (see section 2).

Appendix C

Dr Juan Carlos Cisneros
Museu de Arqueologia e Paleontologia
Universidade Federal do Piauí
Teresina, Brazil
November 2nd 2021

Royal Society Open Science
Dear Editor(s)

Response letter

On behalf of my co-authors, I am pleased to submit the revised version of manuscript RSOS-210898, "Digging deeper into colonial palaeontological practices in modern day Mexico and Brazil". All authors listed on the title page have read the revised manuscript, attest to the validity and legitimacy of the data and its interpretation, and agree to its submission for *Royal Society Open Science*.

We are sincerely grateful for the constructive critiques and positive responses from both reviewers. We have considered all of them, and have made revisions to the manuscript in accordance.

We all believe that the revised version is a substantial improvement. Please find our direct responses (demarcated with plain text) to each of the reviewers comments (in bold italicised text) below. We have modified the manuscript according to all the recommendations requested by the referees and we have added several minor corrections in order to improve clarity, grammar, and style.

Two major updates have been included to the manuscript:

1. The first one being the revelation that the "*Ubirajara jubatus*" fossil was bought by the Karlsruhe museum in 2009 and the permanent retraction of its publication by the journal *Cretaceous Research*.
2. The second update concerns the repatriation of 36 fossil spiders from the Araripe Basin that had been exported illegally to the USA.

Both are relevant to the topic of our contribution and are addressed in the discussion, and a new figure (Fig. 10) was also included. A number of new references have been incorporated to the manuscript as well. We have also reorganized the manuscript discussing Mexico before Brazil and the title has been slightly changed to reflect this.

Sincerely,

Dr Juan Carlos Cisneros on behalf of all co-authors
Email: juan.cisneros@ufpi.edu.br

Reviewer #1:

General comments

Firstly, I would like to say that I really enjoyed reading this manuscript. The ‘story’ is interesting, and the way it has been presented is gripping. It reads like a novel that one does not want to put down – well done. It is also a very important topic, and clearly the authors have extensive experience in this field and with this issue. This comes across in the way it has been presented. However, I do have some relatively minor comments, and a more substantial one.

Many thanks for your appreciation.

It is highly irregular to only include a description of the methods in the supplementary information. If a literature review is used, it should be described (even if very briefly) in the main text of the document, so that readers have some idea of the basis of the data used and the conclusions drawn.

The original formatting of the manuscript was made in accordance with the guidelines for the new subject focus of the journal “Science, Society and Policy” that provided example manuscripts/articles with the methods sections in the supplementary. We are aware of the importance of the methods for this particular study and hence have moved the methods section to the main text.

And even when I access the supplementary material, all it contains are hyperlinks to a web based repository that then autodownloads a zip file of further tables... This is not how one reports methods, but is rather an admirable way of presenting the data used. Somewhere, at least in the Supplementary information but preferably in the main text, there has to be a written description of the methods used to obtain the data upon which part of this study has been based. This should also include a brief description of the Twitter analytics alluded to in the Acknowledgements.

For example, in Table S1, there is no description of how these papers were found (what search terms, what databases or platforms?), or of the content of this table (what does n/a mean for ‘collecting permit’? Does it mean there was not one, it was not mentioned, or that the work described in that paper did not need a permit? Under ‘Issues’, what do the different numbers mean?). All of this information needs to be described somewhere. In addition to a description of the methods in the text, I would also recommend adding a ‘read me’ sheet to each of the supplemental data files, where you explain what the different columns and numbers etc mean.

All the relevant information regarding how the publications used in this study were selected (including the sources, platforms and databases used) can be found in the methods section, now found in the main text. We include a list of all issues that we looked at along with the corresponding values as provided in Tables S1 and S2. We also address the issue of no permits being reported in the study. The methods used for the Twitter analytics are now described in this section.

On another note, regarding Table S1, I would like to see the titles of each of the publications, or a full reference, so that readers can go and find these different papers more easily.

The edits have been made for all three tables (S1-S3) that list fossils from Mexico and Brazil.

Regarding the other Supplemental tables (I noticed this issue in Table S4 but it may be true for more of them), while I completely acknowledge that English is not the only language of science and is not the primary language of either of these regions or the authors of this paper, I would suggest having the table headings either in English or including their English translations in a 'read me' sheet. I realize that translating the data will be a pain, but if you are translating the main manuscript into Spanish and Portuguese anyway, then you might consider having different sheets in each (relevant) table for each of the three languages? Or simply provide an explanation in the first 'read me' sheet translating key terms (like "Coleção Científica, exposição, etc."). As an example, if the content of this specific column in Table S4 (Tipo) is confined to only two or three types of work, it might be easier for all different language readers to interpret if you rather set it up more like I have done in the screenshot below (I had to use Google translate, so I'm sure the headings are not quite correct). I appreciate that the issue of language is a tricky one, but for your paper to have maximum impact, it would be ideal for everything to be multilingual, or at least easy to interpret.

We have translated the requested terms inside tables S4 and S5 and made the suggested set up with additional columns.

Specific comments:

For these comments, I refer to the page number from the document, and not the PDF page (for example, the abstract/summary is on page 3 which is the 4th page in the PDF. I will report this as Page 3. Also note that the PDF line numbers do not seem to fully correspond with the lines of text, so just estimate which line of text each line number I refer to is indicating.

Page 3 line16: "Common issues documented in these publications" are these issues truly documented by the publications you found (i.e., do the authors of those publications highlight the issues), or do you mean that they are issues that you found in these publications?

These are issues that we identified in these publications. We have rephrased the sentence to highlight this.

Page 3 line 23: "reposited" is not a commonly used word in English. I would suggest replacing with something like stored, housed, or kept. But I am not of this field, so if the term is commonly used in palaeontology then ignore my comment.

This is a common term in palaeontology.

Page 4 line 19-23: I don't entirely understand this statement. I would recommend expanding what you mean by institutionalisation, and why it benefits colonial advancement

We have now rephrased the sentence to provide more information:

“ The development of scientific disciplines, educational programmes, and academic organisations were all products designed to benefit colonial advancement [3] e.g. advancements in geological tools allowed colonial powers to uncover and exploit several natural resources in colonies. “

Page 5 line 30: consider rather using the term "while" here, rather than "whereas" (it isn't really a contrast to the previous phrase)

The suggested edits were made.

Page 5 line 40: Rather use the term "scientific reproducibility"

The suggested edits were made.

Page 6 line 26: Delete "being" (before "the USA")

The suggested edits were made.

Page 6 line 59: Rather change "previously" to "to notify the council beforehand"

The suggested edits were made.

Page 7 line 11-14: Consider making this sentence shorter (e.g., use "theoretically" instead of in theory, remove "in the first place", remove "properly"). This should make it easier to read, but if its still too long, then consider using punctuation or breaking it into two sentences.

The suggested edits were made. We have also rephrased the sentence as below:

“The 1970 Convention promotes international cooperation between countries as a means to protect cultural heritage and is theoretically central to preventing the illicit trafficking of cultural property.”

Page 7 line 14: Delete "thus" before acknowledge

The suggested edits were made.

Page 7 line 19-26: This is a very long sentence. Rephrase to shorten, add punctuation, and consider rewriting as two sentences.

The sentence was rephrased as below:

“Thus, the signatories of the Convention are required to enact national laws and services for the protection of the cultural heritage. They are also expected to take the appropriate measures so that museums and other institutions within their territories are prevented from acquiring illegally exported cultural property from another country, as well as to cooperate with restitution of the object(s). ”

Page 7 line 42: Is this not meant to be northEASTERN, rather than northwestern Mexico?

That 's correct. We corrected our mistake.

Page 8 line 14: this is the first time the literature review is mentioned. I would strongly suggest that the methods should be mentioned earlier so that the reader understands how you found these papers (refer to my General comments, above)

The methods section has now been moved to the main text and includes information regarding the literature review.

Page 8 line 18: It is not clear how Box 3 relates to this sentence

We removed the citation of Box 3 here. Thanks for pointing this out.

Page 8 line 19: Rather than “The majority of this output, I would rather state it something like “Most of this foreign-led research, ...” to make it clear that “this output” isn’t referring to the 128 papers

The suggested edits were made.

Page 10 line 59: Here again, there needs to be some sort of description of the methods used to find these papers

The methods have been moved to the main text.

Page 11 line 4: Does “them” (the 57.14%) refer to the foreign-led papers? Or of all the papers?

That 's correct. We have made edits so that this is now more clear:

“...and over half of foreign-led publications (57.14%) showed no evidence....”

Page 12 line 24-26: I am not sure whether the evidence can prove that the foreign researches avoided interaction. I would recommend moderating the language. One could say they neglected to include local scientists, or that this displays a disregard for local laws and expertise.

We edited the text as requested:

“They also display a disregard for local expertise to an even greater extent than observed in the examples from Mexico.”

Page 12 line 37: Rather say that they “likely” represent only a small portion. Because you don’t have the other numbers you cannot say for sure that they do represent only a small portion

The suggested edits were made.

Page 14 line 40: do you rather mean “lines of research” or “research practices”?

We mean “research practices”. The suggested edits were made.

Page 14 line 42: “metropolis” isn’t quite the correct word. I think you mean the colonising countries?

The suggested edits were made.

Page 14 line 49: It does not appear to solely be the absence of legislation, seeing as both the case study countries have adequate legislation; it appears to be more about the absence of enforced legislation

We have edited the text to make this more clear:

“The inadequate, or complete absence of, law enforcement aimed at protecting palaeontological resources...”

Page 20 line 14: How does the sale or purchase of fossils differ from the production of stone products (e.g., for paving or kitchen counters) from quarries that have fossil deposits? I am sure I have seen both the above with little fossil structures in them. And many finds occur in these quarries, which I assume are commercial entities? Without these commercial quarries I assume many fossil deposits would not be revealed. I am not an paleontologist at all, so excuse me if this is a very 'lay' question, but I wonder if this is something you could mention/discuss somewhere

We have now added this to our discussion:

“Nevertheless, we do not criticise the commerce of fossil-bearing limestone nor the mining operations per se, as long as they comply with legal requirements and their environmental impact is addressed. Limestone mining is an important source of employment in many areas. Furthermore, many important fossils would not have been uncovered if it wasn't for commercial mining. Local institutions in both Brazil and Mexico regularly visit them in order to prevent significant fossils from being destroyed or accidentally sold as construction material. In Araripe, outreach activities are carried out involving quarry workers in order to raise awareness and encourage them to report these fossils.”

Page 26 line 11: Would it not be possible to develop a framework somewhat akin to PRISMA regarding a checklist of items every paper about fossils should report (e.g., permit information, detailed description of the site, geology etc., current location of the fossils)? And it would be ideal if proposed research protocols were published for peer review before the research is actually undertaken, so that any gaps (e.g., the acquisition of permits) can be highlighted before the study is conducted.

This is an interesting idea, and in fact some journals are actually now requesting permit information in the submission system. The recommendations we provide in our manuscript could be implemented in different ways by the journals.

Page 27 line 23-24: What do you propose as a solution for these illegal fossils? Should they therefore not be studied, or if they are to be studied and published, should the specimens be repatriated or at least moved to public collections? My concern is the consequence if highly valuable, unique specimens fall in this category, then they will not make it to the scientific literature... should the publications simply be open about the fact that they are studying illegal specimens?

We recognize that some illegal fossils are unique and scientifically valuable and their non-study represents a loss to science. One of our points through the manuscript

though, is that science should not be done at the expense of ethics (and laws should not be broken in the way too). An illegal fossil that has been published (and because of this, became public and “discovered” by its source country that previously was not aware of its existence) should be repatriated. This is a moral obligation and a way to legalize it and repair the damage caused to a country’s heritage. But not all the negative effects are fixed with this, because the expertise, techniques, etc. that were developed through its study mostly helped the country in the Global North that illegally acquired the specimen, perpetuating the scientific dependency by a country in the Global South. The issue of dubious data that is inherent to these fossils is not fixed with repatriation either, some valuable information on taphonomy, sedimentology, stratigraphy, geographical provenance, etc. will be permanently lost due to the practices inherent to illicit traffic. Ideally journals should not accept manuscripts based on these specimens, this is necessary to reduce the demand of fossils from the illicit market by museums and collectors and thus break the chain of events that damage the source country and its heritage in several ways (scientifically, economically, culturally, etc.). Thus, we consider that it is preferable that illicit fossils are not studied.

Page 51: In the caption of Figure 2, Frey et al 2017 is not referenced as a number the way all other references have been done.

We corrected this.

Page 53 Figure 4: I recommend making this figure just a bit larger (fit the width of the page) so that the font is more legible. Also make the currently grey font colour black to improve contrast

We have improved the size of the fonts and the figure is slightly larger.

Page 58 Figure 9: See whether it is possible to make this figure bigger, to fit the width of the page. I think the font size will be better then, and more legible

We have improved the size of the fonts and the figure is slightly larger.

Reviewer #2:

A lucid, well written, abundantly documented paper, Very necessary in this moment... Only minor modifications indicated directly in the text.

We thank the reviewer for their kind comments. All the suggestions made by the reviewer in the provided PDF file have been addressed.

Appendix D

January 7th 2022

Second review of the manuscript:

“Digging deeper into colonial palaeontological practices in modern day Mexico and Brazil”

(Cisneros et al.)

Excellent paper, well documented and very necessary in this moment.

Please consider the following purely formal corrections (pages of the manuscript, not the complete proof file in pdf; for example, I consider as page 1, the page 1 of the manuscript which is the page 9 of the complete proof file in pdf):

Page 6, fourth and fifth lines - There are confusions in the references of the figures: *Tetrapodophis amplectus* corresponds to figure 2f and “*Ubirajara jubatus*” corresponds to figure 2a.

Page 11 - Erase **3.1**.

Page 12, seventh line (“line16”) - There is an error in the figure reference which is not figure 5 but figure 4. In the phrase “We found no record of Mexican fossils being stored in foreign collections, except for a study published in 1990 [66] that studied a collection of fossils claimed to have been collected in the 1930s”, replace “that studied” by “on” so avoiding the redundancy of “study” and “studied”.

Page 16, “line 46” - Error on the reference of *Tetrapodophis amplectus* which is not 2a but 2f.

Page 17, second line - Error on the reference of “*Ubirajara jubatus*” which is not 1c but 2a.

Page 57 - Change order considering Box 2 for Mexico and Box 3 for Brazil, to be coherent with the main text.

Captions of figures 1, 2, and 11 – Genera and species names should go in italics.

Jean-Noël Martinez

Instituto de Paleontología
Universidad Nacional de Piura
Perú

Appendix E

Dr Juan Carlos Cisneros
Museu de Arqueologia e Paleontologia
Universidade Federal do Piauí
Teresina, Brazil
January 25th 2022

Royal Society Open Science
Dear Editor(s)

Response letter

On behalf of my co-authors, I am pleased to submit the revised version of manuscript RSOS-210898, "Digging deeper into colonial palaeontological practices in modern day Mexico and Brazil". All authors listed on the title page agree to its submission for *Royal Society Open Science*.

We have incorporated the minor modifications required by reviewer 2. In addition, we updated some references that were in press at the moment of writing and added a new one.

Sincerely,

Dr Juan Carlos Cisneros on behalf of all co-authors
Email: juan.cisneros@ufpi.edu.br